# Cytotoxic CD8+ T cells target citrullinated antigens in rheumatoid arthritis

Jae-Seung Moon [1,2,11], Shady Younis [1,2,3,11], Nitya S. Ramadoss[1,2], Radhika Iyer [1,2], Khushboo Sheth [1,2], Orr Sharpe[1,2], Navin L. Rao[4], Stephane Becart[5], Julie A. Carman[4], Eddie A. James [6], Jane H. Buckner[6], Kevin D. Deane [7], V. Michael Holers[7], Susan M. Goodman [8,9], Laura T. Donlin [8,9], Mark M. Davis [3,10] & William H. Robinson [1,2,3] ✉

The immune mechanisms that mediate synovitis and joint destruction in rheumatoid arthritis (RA) remain poorly defined. Although increased levels of CD8+ T cells have been described in RA, their function in pathogenesis remains unclear. Here we perform single cell transcriptome and T cell receptor (TCR) sequencing of CD8+ T cells derived from anti-citrullinated protein antibodies (ACPA)+ RA blood. We identify GZMB+CD8+ subpopulations containing large clonal lineage expansions that express cytotoxic and tissue homing transcriptional programs, while a GZMK+CD8+ memory subpopulation comprises smaller clonal expansions that express effector T cell transcriptional programs. We demonstrate RA citrullinated autoantigens presented by MHC class I activate RA blood-derived GZMB+CD8+ T cells to expand, express cytotoxic mediators, and mediate killing of target cells. We also demonstrate that these clonally expanded GZMB+CD8+ cells are present in RA synovium. These findings suggest that cytotoxic CD8+ T cells targeting citrullinated antigens contribute to synovitis and joint tissue destruction in ACPA+ RA.

Rheumatoid arthritis (RA) is a systemic autoimmune disease that affects synovial joints[1,2]. The majority of RA patients possess anti-citrullinated protein antibodies (ACPAs)[3,4]. The citrullinated epitopes targeted by ACPA arise from the post-translational conversion of peptidyl-arginine to peptidyl-citrulline by peptidyl-larginine deiminases (PADs)[5]. The detection of ACPA in serum using the anti-cyclic citrullinated peptide (CCP) test, is a well-established component of the diagnostic criteria for seropositive RA[6]. There are reports describing both pathogenic[7–9] and protective[10] functions for ACPA. In addition, CD4+ T cells that target citrullinated epitopes presented by MHC class II have been

identified in RA[11–15]. There have also been reports describing activated CD8+ T cells in RA[16,17], but the specificity and role of CD8+ T cells in RA remain poorly understood. There is great need to further define the activity of CD8+ T cells and the mechanisms underlying joint tissue destruction in RA.

The association between RA and specific HLA class II alleles is well established. In particular, HLA-DRB1*04:01 is one of the strongest genetic associations with ACPA+ RA[18,19], and this and related DRB1 alleles facilitate presentation of citrullinated peptides to CD4+ T cells[20,21]. Using HLA-DRB1*04:01:citrullinated-peptide tetramers, CD4+ T cells reactive with citrulline-modified peptides derived from

[1]Division of Immunology and Rheumatology, Department of Medicine, Stanford University School of Medicine, Stanford, CA 94305, USA. [2]VA Palo Alto Health Care System, Palo Alto, CA 94304, USA. [3]Institute for Immunity, Transplantation and Infection, Stanford University, Stanford, CA, USA. [4]Immunology Discovery, Janssen Research and Development LLC, Spring House, PA 19477, USA. [5]Immunology Discovery, Janssen Research and Development LLC, San Diego, CA 92121, USA. [6]Center for Translational Immunology, Benaroya Research Institute, Seattle, WA 98101, USA. [7]Division of Rheumatology, University of Colorado Anschutz Medical Campus, Aurora, CO 80045, USA. [8]Hospital for Special Surgery, New York, NY 10021, USA. [9]Weill Cornell Medicine, New York, NY 10021, USA. [10]Department of Microbiology and Immunology, Stanford University, Stanford, CA 94305, USA. [11]These authors contributed equally: Jae-Seung Moon, Shady Younis. ✉e-mail: w.robinson@stanford.edu

vimentin, aggrecan, and other proteins have been identified in the blood of ACPA+ RA patients[11–15,22].

There are also MHC class I alleles that correlate with susceptibility to RA[23]. A single amino acid polymorphism in the HLA-B8 peptide-binding groove, HLA-B*8-Asp9, is associated with a two-fold increased risk for ACPA+ RA and is also associated with increased risk for myasthenia gravis and systemic lupus erythematosus[23]. Association of the HLA class I allele with seropositive RA and other autoimmune diseases suggests that CD8+ T cells contribute to the pathogenesis of autoimmune diseases[24]. Additionally, there is growing evidence that CD8+ T cells recognizing self-peptides may contribute to a variety of autoimmune tissue damages in mice and humans[25–27].

CD8+ T cells that express activation markers are abundant in the joints of RA patients[16,17,28]. Of note, distinct subsets of CD8+ T cells that express *GZMK*, *GZMB* and/or *GNLY* are detected in the synovium of patients with RA[29,30]. However, it remains unclear whether these populations contain clonally expanded CD8+ lineages and their antigen targets remain undefined. CD8+ T cells derived from RA patient blood are known to exhibit an increased proportion of effector subsets (CD27−CD62L−) that express pro-inflammatory cytokines and granzymes compared to healthy controls[16,31,32]. Further, CD8+ T cells in RA blood exhibit oligoclonal T cell receptor (TCR) expression and interestingly, immune checkpoint inhibitor-induced arthritis patients also have an expanded cytotoxic CD8+ T cells in the joints[33–37]. Nevertheless, comprehensive phenotypic and TCR analysis of CD8+ T cells in RA has not been described.

In this work, we use flow cytometry, single cell transcriptomics, and functional assays to characterize the properties of CD8+ T cells in RA. We identify expanded clonal lineages of *GZMB*+ and *GZMK*+ *CD8*+ T cells expressing cytotoxic, pro-inflammatory and tissue homing transcriptional programs in the blood of ACPA+ RA patients. The dominant large expanded *GZMB*+ clonal lineages that highly co-express *GNLY* are also present in RA synovium. We show that stimulation of ACPA+ RA CD8+ T cells with citrullinated antigens in the context of HLA class I results in their proliferation, clonal expansion, expression of cytotoxic mediators and chemokine receptors, and activation to kill target cells. Together, our findings show that cytotoxic CD8+ T cells targeting citrullinated proteins may contribute to synovitis and joint tissue destruction.

## Results

### CD8+ T cells in ACPA+ RA blood express activation and cytotoxic markers

We first performed flow cytometry analysis of peripheral blood mononuclear cells (PBMCs) from ACPA+ RA patients, ACPA− RA patients, and healthy controls (HC) (Supplementary Fig. 1 and Supplementary Table 1). We found that the percentage of CD8+ T cells is increased in the blood of ACPA+ RA as compared to ACPA− RA patients or HCs (Fig. 1a) and is positively correlated with serum anti-CCP antibody levels (Fig. 1b). We detected CD8+ T cells in ACPA+ RA blood highly express the activation and cytotoxic markers CD69 and GPR56 (Fig. 1c, d), and showed reduced expression of the inhibitory receptors PD-1 and TIM3 as compared to HCs (Supplementary Fig. 2a). The CD8+TCRγδ+ T cell population was significantly increased in both ACPA+ and ACPA− RA as compared to HC blood (Fig. 1e). Moreover, the frequency of CD8+ T cells that express granzyme B (GzmB) in ACPA+ RA blood was significantly increased as compared to HCs, while the proportion of granzyme K (GzmK)+CD8+ T cells was similar between the groups (Fig. 1f).

We next compared memory CD8+ T cell subsets, including naïve (N), central memory (CM), effector memory (EM), and activated effector memory CD45RA+ (EMRA) based on the expression of CCR7 and CD45RA. CD8+ T cells in both ACPA+ and ACPA− RA exhibited a lower proportion of naïve T cells and increased levels of EMRA cells when compared to HCs (Fig. 1g). The proportion of CD8+ EMRA cells that express KIR2DL3 which is a marker of regulatory CD8+ T cells[38] and

CD69 was significantly increased in both ACPA+ and ACPA− RA as compared to HCs (Supplementary Fig. 2b, c). Further, we found that not only the proportion of total CD8+ T cells, GzmB or KIR2DL3-expressing CD8+ T cells or EMRA CD8+ T cell subsets, but also the absolute cell counts of these activated cytotoxic CD8+ cells, are increased in ACPA+ RA blood as compared to HC (Supplementary Fig. 3). Together, these results demonstrate that RA blood CD8+ T cells express increased CD69, GPR56 and granzyme B, and are skewed to an activated effector memory (CD45RA+ CCR7−) phenotype.

### Single-cell transcriptomics identifies multiple activated cytotoxic CD8+ T cell subpopulations in ACPA+ RA as compared to HCs

For in-depth characterization of CD8+ T cells in RA, we performed single-cell transcriptome (and TCR) sequencing of CD8+ T cells in the blood of ACPA+ RA and HCs. We analyzed the whole transcriptome datasets to select CD8+ T cells using unsupervised clustering of total CD3+ T cells to define the CD8+ T cell clusters based on the expression of CD8A protein using CITE-seq antibodies and RNA levels of *CD8A* and *CD4* (*CD4* < 1) (Supplementary Fig. 4a). This analysis resulted in the identification of 7 distinct clusters of CD8+ T cells using unsupervised clustering. We annotated these clusters based on their expression of canonical subset markers, and defined the following clusters of CD8+ T cells in ACPA+ RA and HC blood: Naïve, Memory, *TCRgd*+, *GZMK*+, *GZMB*+ *GNLY*+, *GZMB*+ *KIR*+ and *CCR6*+ *CD161*+ (Fig. 2a, b, Supplementary Fig. 4b and Supplementary Table 2). These subpopulations were transcriptionally distinct based on the analysis of differentially expressed genes (adj.P value < 0.05; log2 fold change > 0.8) (Supplementary Fig. 4c). Interestingly, we detected *KIR3DL2*, *KIR2DL3* or *KIR3DL1*-expressing CD8+ T cells express several cytotoxic markers including *GZMB*, *GNLY* or *PRF1* as previously published[38].

We analyzed differences in the distribution of CD8+ T cells across the 7 identified clusters. ACPA+ RA patients showed an increase in the proportion of multiple clusters including *GZMB*+ *KIR*+, *TCRgd*+ and, while the proportion of CD8+ T cells in the *CCR6*+ *CD161*+ cluster was decreased in ACPA+ RA (Fig. 2c).

We next analyzed the populations of granzyme-expressing CD8+ T cells between ACPA+ RA and HCs using pseudobulk differential expressed gene (DEG) analysis. We performed gene set enrichment analysis (GSEA) to identify the pathways represented by the DEGs. The DEGs expressed by the *GZMB*+ clusters from ACPA+ RA represented pathways including T cell mediated cytotoxicity, leukocyte mediated cytotoxicity, and cell killing; while in contrast the DEGs expressed by the *GZMK*+ clusters from ACPA+ RAs represented pathways including T cell receptor signaling, and gamma delta T cell differentiation (Fig. 2d). In addition, the GO enrichment analysis indicated a significant enrichment of T cell cytotoxicity and cell killing pathways in ACPA+ RA *GZMB*+ cluster (adj.P value < 0.05) compared to HC *GZMB*+ cluster, while the ACPA+ RA *GZMK*+ cluster was non-significant between ACPA+ RA and HC (Fig. 2e).

Analysis of the granzyme-expressing clusters (*GZMB*+ and *GZMK*+), demonstrated that *GZMB*+ clusters express significantly increased levels of the T cell cytotoxicity-related genes *GNLY*, *PRF1, CTSC* and *CD226* in ACPA+ RA as compared to HCs, while the expression level of these genes is similar in *GZMK*+ cluster between ACPA+ RA and HC (Fig. 2f, g). Interestingly, the pseudobulk expression level of all these genes was higher in the *GZMB*+ clusters as compared to *GZMK*+ cluster in ACPA+ RA. These results demonstrate prominent *GZMB*-expressing CD8+ T cell subpopulations express transcriptional programs of T cell cytotoxicity in ACPA+ RA as compared to HCs.

### Identification of clonally expanded CD8+ lineages in ACPA+ RA as compared to HCs

We further performed integrated analysis of the single-cell transcriptomic and TCR sequencing data, to analyze clonal lineages

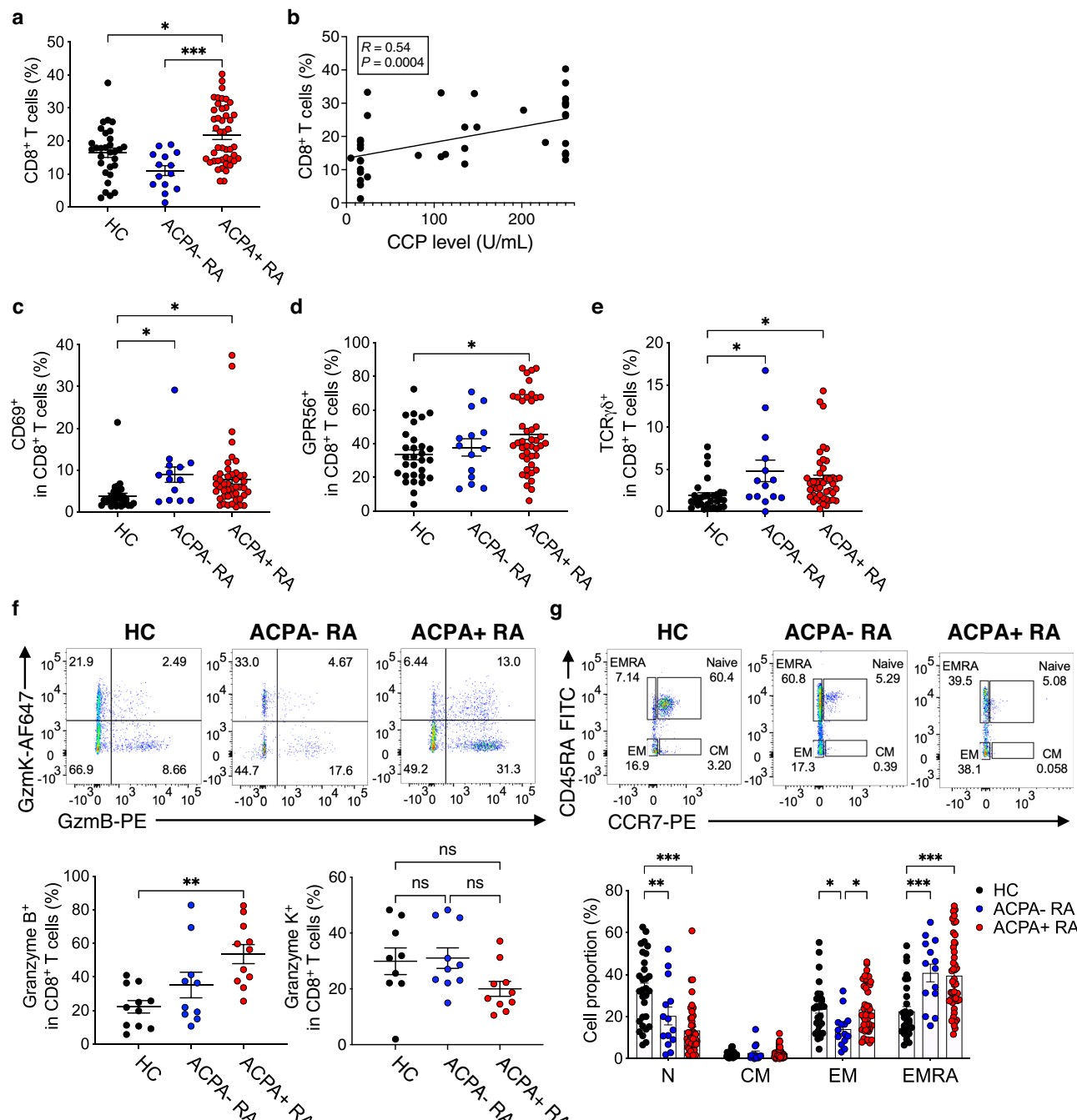

**Fig. 1 | Increased levels of CD8+ T cells expressing cytotoxic mediators in the blood of ACPA+ RA patients. a** Frequency of CD8+ T cells among total lymphocytes in PBMCs from healthy controls (HC; *n* = 30), ACPA− RA (*n* = 14), and ACPA+ RA (*n* = 45) measured by flow cytometry (*P* = 0.0179, or ***P* = 0.0001). **b** Spearman's correlation between the ratio of CD8+ T cells and serum anti-CCP antibody (surrogate for ACPA) levels in RA (*n* = 39; *R* = 0.5434, *P* = 0.0004). **c**−**e** Percentage of CD69, GPR56, or TCRγδ-expressing CD8+ T cells: HC (*n* = 30), ACPA− RA (*n* = 14), and ACPA+ RA (*n* = 45). For **c**, *P* = 0.0355, or *P* = 0.0233. For **d**, *P* = 0.0231. For **e**, *P* = 0.0120, or *P* = 0.0237. **f** Representative flow cytometry results (*upper*) and quantified graphs (*lower*) of GzmB, or GzmK expressing CD8+ T cells in HC (*n* = 9 for GzmK, or 11 for GzmB) and ACPA+ RA (*n* = 10 for GzmK, or 11 for GzmB). For

GzmB+CD8+ T cells, **P* = 0.0014. **g** Flow cytometry analysis of memory subsets of CD8+ T cells: CCR7hiCD45RAhi (Naïve), CCR7hiCD45RAlow (CM, Central memory), CCR7lowCD45RAlow (EM, Effector memory), CCR7lowCD45RAhi (EMRA, Effector Memory cells Re-expressing CD45RA) (*upper*). Quantified proportion of each memory subset in HC (*n* = 30), ACPA− RA (*n* = 14), or ACPA+ RA (*n* = 45) patients (*lower*). In N, **P* = 0.004, or ***P* < 0.001. In EM, *P* = 0.03. In EMRA, ***P* < 0.001. Data are presented as means ± SEM. *P* < 0.05, ***P* < 0.01, or ****P* < 0.001 by ordinary one-way ANOVA (**a**, **c**–**f**), and two-way ANOVA (**g**) with Tukey's multiple comparisons test. ns, not significant. Source data are provided as a Source Data file. ACPA anti-citrullinated protein antibodies, RA rheumatoid arthritis, CCP cyclic citrullinated peptide, GzmB Granzyme B, GzmK Granzyme K.

within the 7 CD8+ T cell clusters. We defined expanded clonotypes based on two or more T cells expressing the same paired *TCRα* and *TCRβ* chains. From the ~10,000 *TCRαβ* pairs obtained, we identified a range of clonal lineage sizes including Large (clonal lineage members, X > 20), Medium (20 > X ≧ 5), and Small (5 > X ≧ 2)

lineages. CD8+ T cells in ACPA+ RA contained a higher proportion of clonally expanded cells than HC (Fig. 3a), suggesting CD8+ T cells could be clonally expanded in response to specific antigen(s) in RA. The clonally expanded CD8+ T cells were mostly located in effector/cytotoxic clusters, including the *GZMB+ GNLY+*, *GZMB+ KIR+*, *TCRgd+*,

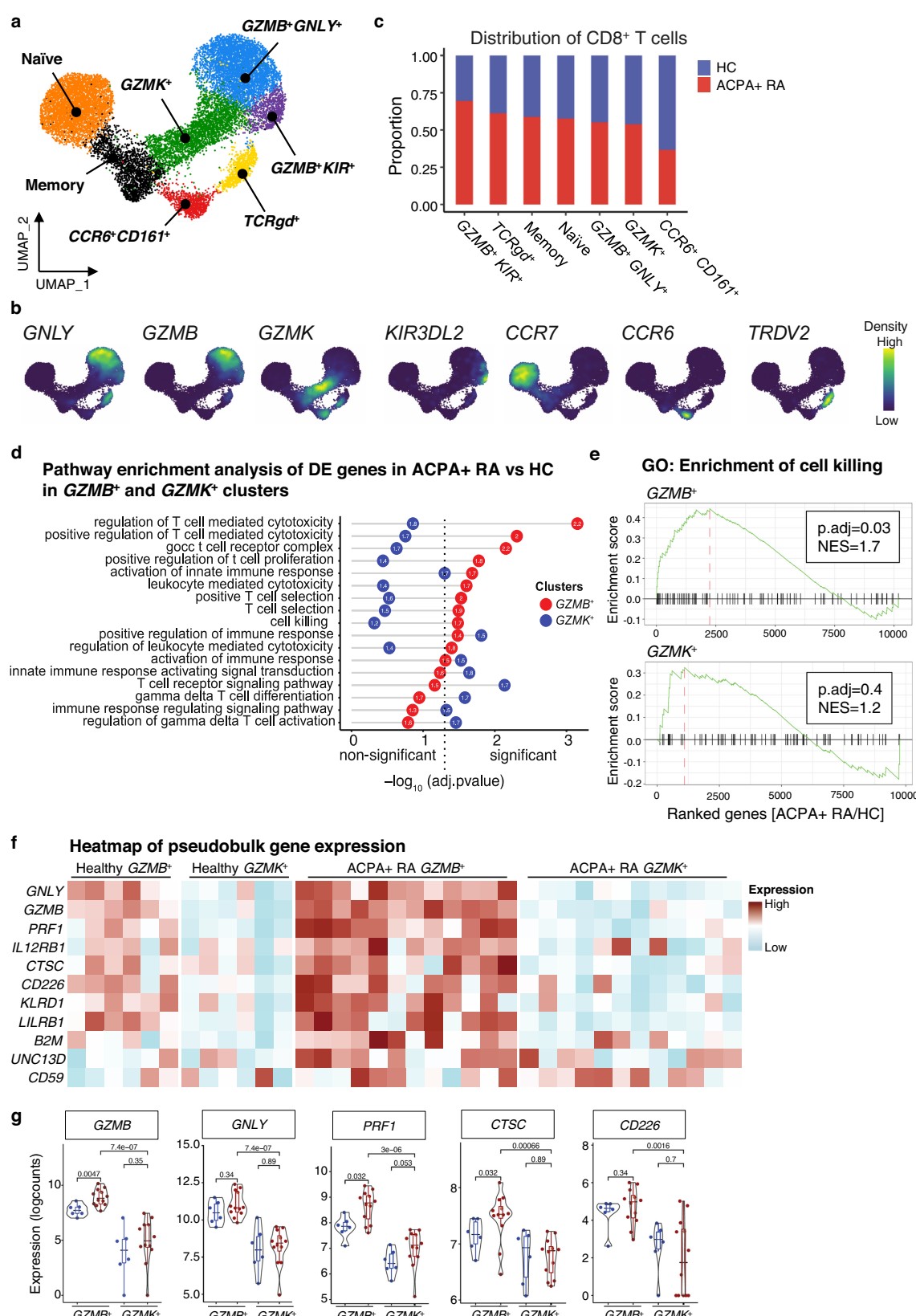

and *GZMK*[+] clusters (Fig. 3b). In contrast, we observed naïve and memory clusters predominantly consist of singletons (Supplementary Fig. 5a, b). The proportion of largely expanded *GZMB*[+] *GNLY*[+] cells was considerably higher in ACPA+ RA (~50% of total clones) than in HCs (~30% of total clones), suggesting *GZMB* and *GNLY*-expressing CD8[+] T cells could represent cytotoxic cells

responding to an autoantigen (Fig. 3c). In contrast, the *GZMK*[+] clusters exhibited a relatively lower proportion of expanded clones as compared to the *GZMB*[+] clusters in ACPA+RA (Fig. 3c). In addition to the proportion, the absolute frequency of largely expanded clones in *GZMB*[+] *GNLY*[+] cluster was higher than in *GZMK*[+] cluster in ACPA+ RA (Fig. 3d).

**Fig. 2 | Transcriptional landscape of CD8+ T cells in ACPA+ RA and HC blood.** Single cell RNA-seq of CD8+ T cells in ACPA+ RA ($n = 12$) and healthy ($n = 6$) PBMC samples using the 10X Genomics Chromium platform. **a** UMAP plot of 7 annotated clusters from CD8+ T cells ($n = 16,000$) in ACPA+ RA and healthy PBMC samples. **b** Density of mRNA expression of the key markers for CD8+ T cells clusters. **c** Distribution of CD8+ T cells from ACPA+ RA and HC in each cluster. **d** Dot plot of enrichment analysis of pseudobulk differentially expressed genes (DEGs) between *GZMB*+ clusters in ACPA+ RA and HC cells (red) or *GZMK*+ clusters in ACPA+ RA and HC cells (blue). The calculated log2 fold-change of gene expression in ACPA+ RA vs HC were multiplied by the -log (FDR) and the output values were used to rank the genes descending and perform gene-set enrichment analysis (GSEA) in *GZMB*+ or *GZMK*+ clusters independently. Numbers in the circle indicate the normalized enrichment score (NES) for each pathway in *GZMB*+ (red) or *GZMK*+ (blue). Adjusted *P* value (p.adj) for multiple testing using Benjamini–Hochberg. **e** Enrichment GO analysis of cell killing pathway of upregulated genes in *GZMB*+ (ACPA+ RA vs HC) or *GZMK*+ (ACPA+ RA vs HC). **f** Heatmap of normalized pseudobulk expression of genes involved in the cell killing and T cell cytotoxicity enrichment pathway. Each column represents one pseudobulk expression of one sample of the indicated clusters. **g** The direct comparison of key cytotoxic genes and their differential expressions across *GZMB*+ vs *GZMK*+ clusters in ACPA+ RA ($n = 12$) and HC ($n = 6$) CD8+ T cells. *P* values of Wilcox test are presented on the top of each comparison. The boxplot represents the median shown as a line in the center of the box, the boundaries are the first and third quartile, and whiskers represent the minimum and maximum values in the data. Source data are provided as a Source Data file. HC healthy control, ACPA anti-citrullinated protein antibodies, RA rheumatoid arthritis, DE differentially expressed, GO gene ontology.

Further, we observed clonally expanded CD8+ T cells expressed cytotoxic mediators including *GZMA*, *GZMB*, *GNLY*, and *PRF1*, as well as cell migratory markers including *CX3CR1* and *CCL4* (Fig. 3e). Extending from our flow cytometric data (Fig. 1f), we demonstrated that the level of *GZMB* expression is increased in CD8+ T cells of ACPA+ RA as compared to HCs, even in the large clonally expanded cells. On the other hand, the strong expression of *GZMK* was detected in medium and small expanded clones or some singletons. Singletons don't express cytotoxic and migratory genes, suggesting the population of antigen-responsive CD8+ T cells highly expresses cell-killing genes. Moreover, analysis of TCR Vβ and Jβ gene expression showed increased frequency of expression of certain *TRBV* and *TRBJ* genes, including *TRBV5-4–TRBJ2-3* and *TRBV29-1–TRBJ2-7* in ACPA+ RA compared to HCs (Supplementary Fig. 5c, d). Together, these results indicate clonally expanded CD8+ T cells in ACPA+ RA blood express genes encoding cytotoxic mediators, including *GZMB*.

**Citrullinated antigens stimulate ACPA+ RA blood CD8+ T cells to proliferate in an HLA class I-dependent manner**

Given the increased proportions and clonal expansions of granzyme B-expressing CD8+ T cells in ACPA+ RA blood, we investigated whether ACPA+ RA CD8+ T cells are reactive to candidate citrullinated autoantigens. We first measured levels of the Ki-67 proliferation marker on ACPA+ RA PBMCs stimulated with citrullinated vimentin (cit-vimentin) or native vimentin protein. Cit-vimentin is a citrullinated protein present in the RA joint[22,39]. We used anti-CD3/CD28 antibodies, influenza virus nuclear protein (NP), and/or cytomegalovirus (CMV) pp65 as positive controls. Stimulation with cit-vimentin, but not native-vimentin, significantly increased the frequency of Ki-67+ cells among CD8+ T cells (Fig. 4a). Further, antibody blockade of HLA class I-mediated antigen presentation using anti-CD8 or HLA class I (HLA I) antibody inhibited cit-vimentin-induced Ki-67 expression (Fig. 4a), indicating HLA class I-restricted presentation of cit-vimentin mediated CD8+ T cell activation. We confirmed both cit-vimentin and influenza NP were endocytosed by HLA class I-expressing CD11c+ antigen-presenting cells (APCs) (Supplementary Fig. 6a, b). In a parallel experiment, cit-vimentin but not native vimentin also stimulated ACPA+ RA CD4+ T cells to express Ki-67 (Fig. 4b).

To further characterize RA CD8+ T cell responses to cit-vimentin, ACPA+ RA CD3+ T cells were stained with a cell proliferation dye and cocultured with monocyte-derived dendritic cells (MoDCs) loaded with cit-vimentin or native vimentin. Cit-vimentin induced significant proliferation of both CD8+ and CD4+ T cells as measured by increased dye^low CD8+ or CD4+ T cells, while native vimentin did not induce proliferation, and the antibody-mediated blockade of HLA class I-mediated antigen presentation blocked the proliferation (Fig. 4c, d). To determine if cit-vimentin can directly activate CD8+ T cells to proliferate without CD4+ T cell help, CD8+ T cells were isolated and cocultured with cit- or native vimentin. CD8+ T cells proliferated in response to cit-vimentin, and blockade of HLA class I significantly inhibited cit-vimentin-induced proliferation (Fig. 4e). In contrast, cit-

vimentin did not induce proliferation of ACPA− RA or HC CD8+ T cells (Supplementary Fig. 7a–c). Collectively, these results show CD8+ T cells in ACPA+ RA blood exhibit proliferative responses to cit-vimentin in an HLA class I-restricted manner.

**Citrulline-reactive CD8+ T cells in ACPA+ RA are clonally expanded and highly express cytotoxic and synovium-trafficking molecules**

To define whether CD8+ T cells undergo clonal expansion in response to citrullinated antigens, we used single cell RNA and TCR sequencing. We performed antigen stimulation of blood CD4+ and CD8+ T cells from 3 ACPA+ RA patients with native/ citrullinated vimentin or H3 proteins as described in Fig. 4c, d. We identified 17 distinct clusters of CD4+ and CD8+ T cells, with clusters 3, 6, 10 and 14 being composed of CD8+ T cells that express granzymes (Supplementary Fig. 8a–c). Most of large and medium clonally expanded cells expressed *CD8* and not *CD4* (Supplementary Fig. 9a). Further, *CD8*+ clonal lineages expressed cytotoxic and chemokine molecules (CD57, *GZMH*, *GNLY*, *GZMB* and *CCL4*), while the *CD4*+ cells represented smaller clonal lineages and singletons expressing naïve phenotypes (CD62L, CCR7 and IL-7Ra) (Supplementary Fig. 9b).

We showed citrullinated antigens stimulate clonal expansion of T cells (Supplementary Fig. 10a). While most RA CD3+ T cells exhibited high inflammation scores, cytotoxicity and proliferation scores were only increased by CD8+ T cells contained in the citrulline-reactive clusters 3, 6, 10 and 14 (Supplementary Fig. 10b). Next, we isolated CD8^high CD4^low cells and re-clustered for in-depth analysis of citrulline-reactive CD8+ T cells (Supplementary Fig. 11a). Within the CD8+ T cell clusters, citrullinated antigens induced large cluster 0 clonal lineages expressing high cytotoxic and synovium-trafficking scores (Fig. 5a–c). Using CD45RO and CD45RA CITE-seq antibodies, we identified the citrulline-reactive cluster 0 shows a phenotype of activated memory or TEMRA subset (Supplementary Fig. 11b, c). The majority of the citrullinated vimentin or H3-reactive expanded clones was located in cluster 0, and the clonally expanded cells expressed high levels of *GZMB*, *IFNG* and *MKI67* (Fig. 5d–f), consistent with findings in Fig. 3.

**Identification of RA synovial cells with high transcriptional similarity to citrullinated antigen-reactive blood CD8+ T cells**

To investigate whether the citrullinated antigen-reactive clones identified in ACPA+ RA blood might also be present in the synovium, we performed single-cell RNA-seq analysis of 4 synovial tissues and 3 matched blood samples from ACPA+ RA patients. We identified clonally expanded populations of blood and synovial cytotoxic CD8+ T cells among total CD3+ T cells with the large (clonal lineage members, $1 \geq X > 0.01$), medium ($0.01 \geq X > 0.002$), and small ($0.002 \geq X > 0.001$) or singleton ($0.001 \geq X > 0$) (Supplementary Fig. 12a, b). Most of the clonally expanded cells were in the same-population of blood and synovial tissues and these clonally expanded populations of blood and synovial CD8+ T cells expressed

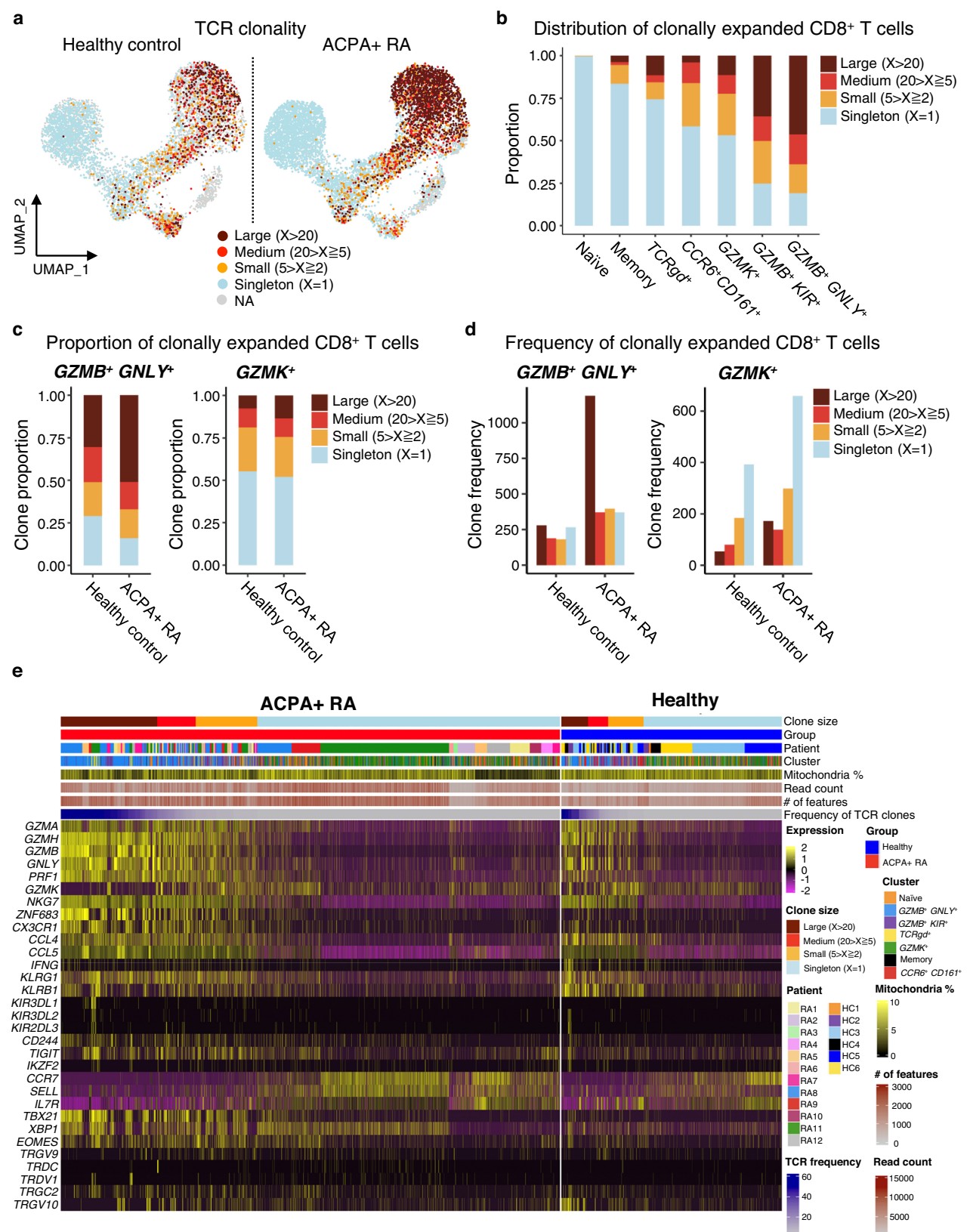

high levels of *GZMB* and *GNLY*, defining the expanded citrullinated antigen-reactive cytotoxic CD8+ lineages in RA blood (Supplementary Fig. 12c). In contrast, most *GZMK*+ cells in RA blood and synovium were in distinct populations with the *GZMB*+ *GNLY*+ cells even though some of *GZMK*+ cells were also expressing *GZMB* (Supplementary Fig. 12c).

We next used the paired TCR analysis to determine whether the shared clonotypes across blood and synovium exhibit a similar cytotoxic transcriptomic phenotype, including *GZMB* or *GZMK* expression. We selected the top synovial tissue clonotypes from two representative ACPA+ RA patients and matched the clonotypes to blood from the same patients. As shown in Fig. 5g, we found the

**Fig. 3 | Single-cell transcriptomics and TCR sequencing demonstrate clonally expanded CD8+ T cells expressing cytotoxic mediators in ACPA+ RA blood.** **a** UMAP plots of healthy ($n = 6$) or ACPA+ RA ($n = 12$) CD8+ T cells integrated with TCR clonality ($n = 10,400$ paired *TCRαβ* sequences). Color indicates the groups by the frequency of clonotypes in total cells. X represents the frequency of each clonotype defined by its unique paired *TCRαβ* sequence. Large-expanded clones ($X > 20$), Medium-expanded clones ($20 > X \geqq 5$), Small expanded clones ($5 > X \geqq 2$), and Singletons ($X = 1$). Cells lacking *TCRαβ* sequence information were designated as NA (Not Applicable in this analysis). **b** Proportion of clonal lineages based on clonal size in each cluster. **c** Bar plots comparing the percentage of clonally expanded cells in *GZMB* (left) or *GZMK* (right)-expressing clusters between ACPA+ RA and HC. **d** Bar plots of absolute numbers of T cell clones in *GZMB+* and *GZMK+* clusters. **e** Heatmap of select gene markers and pathways expressed by clonally expanded and singleton CD8+ T cells in ACPA+ RA and HCs. Top bars indicate clonal size, patients /disease state, clusters, patient sample ID and quality control metric. Individual T cells are ordered by clonal size. Source data are provided as a Source Data file. HC healthy control, ACPA anti-citrullinated protein antibodies, RA rheumatoid arthritis, TCR T cell receptor.

most largely expanded clonotypes from synovial tissues are also present in the matched blood in both patients. Further, we showed these clonally expanded CD8+ T cells in blood and synovium exhibited high similarity of transcriptional programs, including expression of the cytotoxic markers *GZMB*, *GZMK* or *GNLY* (Fig. 5h). Interestingly, the expanded clonotypes specifically expressed *GNLY* in the *GZMB+* cells, while *GZMK+* cells exhibited lower expression of *GNLY*, suggesting that *GZMB+* CD8+ T cells have a cytotoxic function in the RA synovium.

We observed that the majority of the shared clonotypes in blood and synovium expressed high levels of *GZMB* and *GNLY* (Supplementary Fig. 13a–c), while a limited number of shared clonotypes co-expressed *GZMB* and *GZMK* but not *GNLY* (Supplementary Fig. 13d). Thus, these results suggest the citrullinated antigen-responsive cytotoxic *GZMB+ GNLY+* clonotypes in blood are also present in ACPA+ RA synovium.

### Citrullinated antigens activate ACPA+ RA CD8+ T cells to produce cytotoxic mediators

We next sought to examine the cytotoxic capacity of citrullinated antigen-stimulated ACPA+ RA CD8+ T cells. We stimulated fresh whole blood from ACPA+ RA or HC with cit-vimentin or native vimentin for 6 h, and measured the expression of CD69, GzmB and IFNγ in both CD8+ and CD4+ T cells by flow cytometry. Cit-vimentin stimulation significantly increased the proportion of CD69+CD8+ and GzmB+IFNγ+CD8+ T cells from ACPA+ RA blood as compared to no treatment or native vimentin stimulation (Fig. 6a). In contrast, HC blood did not exhibit reactivity to cit-vimentin. Both ACPA+ RA and HC bloods exhibited CD8+ T cell responses to viral proteins (Fig. 6a). Similar results were observed for IFNγ-producing CD4+ T cells (Fig. 6a).

We evaluated whether other citrullinated antigens induce activation of CD8+ T cells in ACPA+ RA PBMCs. We stimulated ACPA+ RA PBMCs with 100 μM of individual citrullinated proteins including cit-vimentin, cit-fibrinogen, cit-α-enolase, cit-H2B, cit-H3 or cit-H4; or a mix of all 6 citrullinated proteins each at 20 μM; or a mix of all 6 native proteins each at 20 μM; and measured IFNγ and GzmB expression by CD8+ T cells by flow cytometry. Cit-vimentin, cit-α-enolase and cit-H3 all significantly stimulated ACPA+ RA CD8+ T cells to express IFNγ and/or GzmB, as compared with stimulation with the native proteins or no treatment (Fig. 6b and Supplementary Fig. 14a).

We showed the frequency of IFNγ+ and GzmB+IFNγ+ CD8+ T cells increased in response to cit-vimentin in a concentration-dependent manner, while native vimentin or no treatment did not activate such responses (Fig. 6c and Supplementary Fig. 14b). Antibody-mediated blocking of the CD8-HLA class I complex inhibited citrullinated antigen mediated stimulation of IFNγ and/or GzmB expression by CD8+ T cells (Fig. 6d and Supplementary Fig. 14c). In contrast, ACPA− RA and HC CD8+ T cells were not activated by these citrullinated proteins (Supplementary Fig. 15a, b). Stimulation with cit-H3 induced ACPA+ RA CD8+ T cells to express cytotoxic mediators and blockade of CD8 or the HLA class I complex inhibited this response (Supplementary Fig. 16), demonstrating activation of anti-cit-H3 CD8+ T cells is also dependent on HLA class I presentation of cit-H3.

### Citrullinated antigens stimulate ACPA+ RA CD8+ T cells to mediate cytotoxic activity

To determine whether citrulline-reactive CD8+ T cells-expressing cytotoxic mediators mediate cytolytic activity, we measured the expression of CD107a, a marker expressed during secretion of lytic granules. Stimulation of ACPA+ RA CD8+ T cells with cit-vimentin or cit-H3 significantly elicited the degranulation process as measured by CD107a expression (Supplementary Fig. 17a, b). Anti-CD8 and anti-HLA class I blocking antibodies inhibited cit-vimentin or H3-induced CD107a expression (Supplementary Fig. 17a, b). Next, we tested the ability of citrullinated antigens to activate ACPA+ RA CD8+ T cells to mediate cell killing. Cit-vimentin, but not native-vimentin, stimulated ACPA+ RA CD8+ T cells significantly killed DLD-1 tumor cells (Supplementary Fig. 17c). In contrast, HC CD8+ T cells were not activated by cit-vimentin to mediate cytotoxic killing but did mediate killing when stimulated with a mix of viral influenza NP and CMV pp65 (Supplementary Fig. 17d). Together, these results demonstrate that citrullinated antigens can activate ACPA+ RA CD8+ T cells to mediate cytotoxicity.

## Discussion

Anti-citrullinated protein autoimmunity is a hallmark of RA, yet it has been unclear whether anti-citrullinated protein responses contribute to the pathogenesis of RA. Here, we demonstrated that ACPA+ RA CD8+ T cells are clonally expanded, and exhibit activated and cytotoxic phenotypes. We further showed that ACPA+ RA CD8+ T cells recognize citrullinated antigens in an HLA class I-dependent manner, and are activated by citrullinated antigens to produce cytotoxic mediators and mediate cell killing. Our findings indicate that anti-citrullinated protein cytotoxic CD8+ T cells could mediate joint tissue destruction in RA.

Using single cell whole transcriptome and TCR sequencing, we identified 7 distinct subpopulations of CD8+ T cells in ACPA+ RA, including naïve/memory as well as *GZMB* or *GZMK* expressing subpopulations. As compared to HCs, ACPA+ RA patients exhibited increased levels of clonally expanded CD8+ T cells expressing cytotoxic genes. Further, the *GZMB+* clusters predominantly contained large clonally expanded lineages and exhibited increased expression levels of cytotoxic, pro-inflammatory and tissue-homing transcriptional signatures as compared to the *GZMK+* and naïve/memory clusters.

In seropositive RA, detection of ACPAs via the CCP antibody assay provides a sensitive and specific diagnostic marker[3,8,40]. This suggests that the generation of citrullinated proteins by PADs and/or immune tolerance to these citrullinated proteins is dysregulated in RA. Here we demonstrated that ACPA+ RA patients possess cytotoxic CD8+ T cells specific for citrullinated antigens. We showed that CD8+ T cells in ACPA+ RA blood proliferate in response to citrullinated, but not native, vimentin and histone H3. Moreover, single cell transcriptome and TCR sequencing defined that citrullinated antigens drive clonal expansion of cytotoxic CD8+ T cells expressing synovium-trafficking molecules. Further, similar to the expanded *GZMB+ GNLY+* CD8+ T cells in RA blood, our paired sample analyses suggest that this population is also expanded in the synovium of matched RA patients and the clonally expanded clone has a shared transcription profile within an individual clonotype across blood and synovium. Our paired TCR analysis confirmed previous findings that ACPA positivity

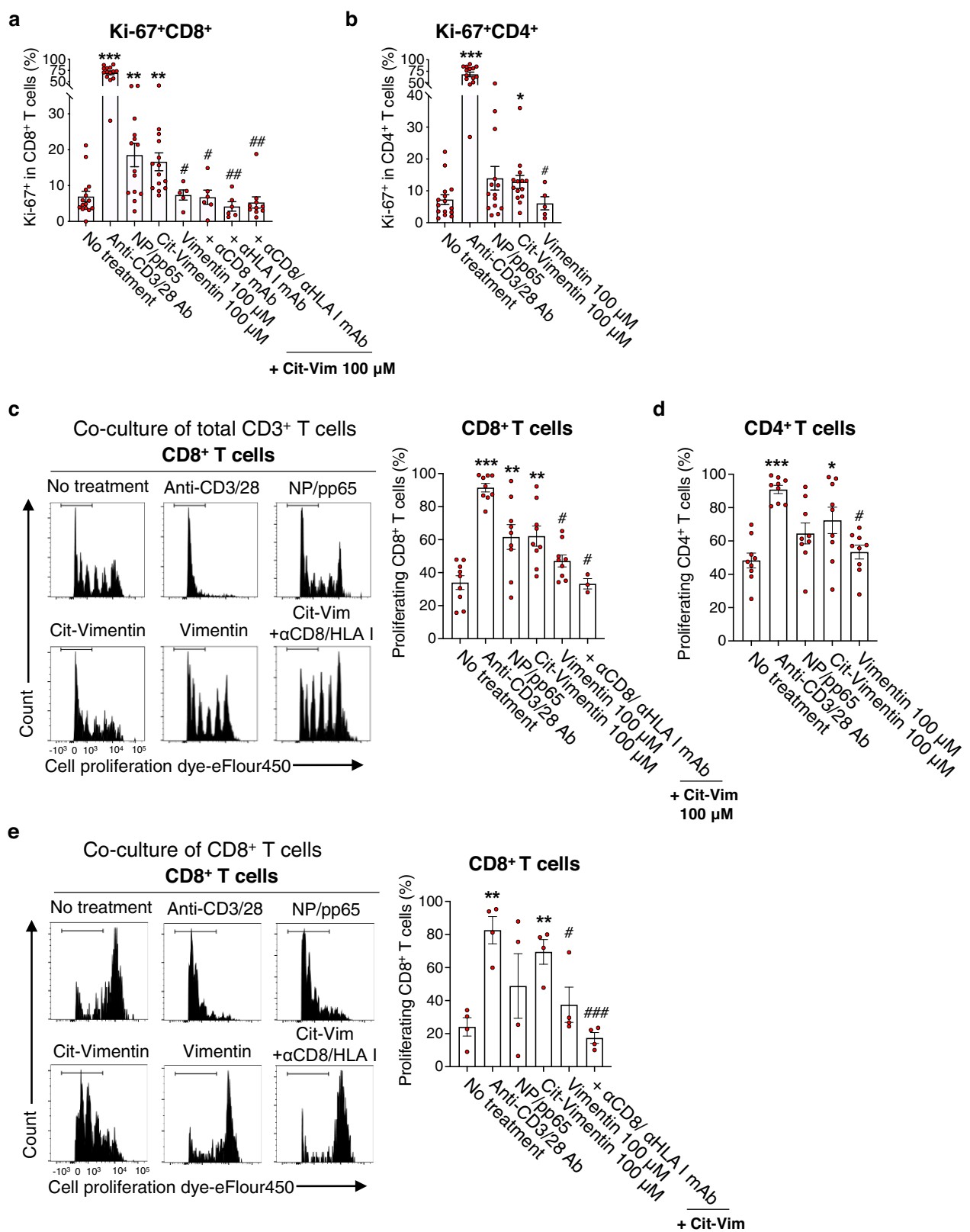

and reactivity to citrullinated antigens are associated with the presence of CD8⁺ T cells in RA synovium[41,42]. To determine whether activation of ACPA+ RA CD8⁺ T cells by citrullinated antigens is mediated by antigen presentation of HLA class I, we show that anti-HLA class I and anti-CD8 blocking antibodies inhibited their activation by citrullinated antigens. These data highlight that

ACPA+ RA patients possess expanded clonal lineages of CD8⁺ T cells that can be activated by citrullinated antigens presented by HLA class I. Given that cit-fibrinogen and certain other citrullinated antigens are extracellular, our findings suggest that citrullinated antigens may be cross-presented in HLA class I to activate cytotoxic CD8⁺ T cells in RA.

**Fig. 4 | CD8⁺ T cells in ACPA+ RA blood proliferate in response to citrullinated antigens. a, b** PBMCs ($5 \times 10^5$) of ACPA+ RA patients were incubated with anti-CD3/28 antibodies ($n = 14$), NP (influenza A)/ pp65 (CMV) protein (50 μM of each, $n = 14$), native vimentin protein (100 μM, $n = 5$), or citrullinated vimentin (cit-vimentin) protein (100 μM, $n = 14$) with or without anti-CD8 and/or HLA class I-blocking antibody (anti-CD8 Ab $n = 6$, anti-HLA class I Ab $n = 6$ or anti-CD8/ HLA class I Abs $n = 10$), or no stimulation ($n = 15$) for 3 days. The percentage of Ki-67-expressing CD8⁺ (**a**) or CD4⁺ (**b**) T cells in ACPA+ RA PBMCs was measured by flow cytometry. For **a**, ***$P < 0.0001$, **$P = 0.026$, ***$P = 0.0021$, #$P = 0.0471$, #$P = 0.026$, ##$P = 0.0056$ or ##$P = 0.0021$. For **b**, ***$P < 0.0001$, or *$P = 0.0383$. **c, d** Native vimentin or cit–vimentin pulsed monocyte-derived dendritic cells (MoDCs) were cocultured with Cell Proliferation Dye eF450-labeled CD3⁺ T cells isolated from ACPA+ RA PBMCs ($n = 9$) with or without anti-CD8/HLA class I-blocking antibody for 10 days, followed by flow cytometry analysis. Representative flow cytometry results (**c**, *left*) and

quantification of the percentage of dye^low proliferating CD8⁺ T cells (**c**, *right*) or CD4⁺ T cells (**d**). For **c**, ***$P < 0.0001$, **$P = 0.0054$, **$P = 0.0015$, #$P = 0.0494$ or #$P = 0.0251$. For **d**, ***$P < 0.0001$, *$P = 0.0171$, or *$P = 0.0484$. **e** Proliferation capacity of ACPA⁺ RA CD8⁺ T cells based on proliferation dye eF450-labeled CD8⁺ T cells co-cultured with cit-vimentin or native vimentin-loaded MoDCs in the absence or presence of anti-CD8/HLA class I-blocking antibody for 10 days ($n = 4$). **$P = 0.0011$, **$P = 0.0027$, #$P = 0.0486$, or ###$P = 0.0007$. Representative histograms (*left*) and quantification of the proliferating CD8⁺ T cells (*right*). Bars represent means ± SEM. *$P < 0.05$, **$P < 0.01$, or ***$P < 0.001$ versus no treatment by unpaired *t*-test with two-tailed test; and #$P < 0.05$, ##$P < 0.01$, or ###$P < 0.001$ versus cit-vimentin treatment by unpaired *t*-test with two-tailed test. Source data are provided as a Source Data file. HC healthy control, ACPA anti-citrullinated protein antibodies, RA rheumatoid arthritis, Cit-Vim citrullinated vimentin.

In addition, ACPA+ RA CD8⁺ T cells expressed cytotoxic and effector molecules in response to stimulation with citrullinated proteins including vimentin, fibrinogen, α-enolase, or histone H3, but not in response to their corresponding native forms. We detected anti-cit-vimentin CD8⁺ T cells using both fresh whole blood as well as PBMCs from ACPA+ RA patients. This finding is important, given such a fresh whole blood assay could provide the basis for a clinical diagnostics to detect anti-cit-vimentin and other citrullinated proteins specific CD8⁺ T cell responses in RA patients – analogous to the QuantiFERON assay that tests for exposure and infection with *Mycobacterium tuberculosis*[43]. Its use as a clinical diagnostic assay for ACPA+ RA requires further validation of specific peptide sequences spanning a citrullinated peptide.

Cytotoxic CD8⁺ T cells produce IFNγ and lytic molecules and have been shown to mediate cell killing and tissue injury in several murine and human autoimmune diseases[25,26,44]. In RA, it is well established that there are increased levels of CD8⁺ T cells in synovium, including tissue-resident memory CD8⁺ T cell populations that potentially contribute to joint destruction by releasing pro-inflammatory cytokines in response to antigens[45]. Here, we showed that ACPA+ RA CD8⁺ T cells mediate cytotoxic responses and kill DLD-1 tumor cells by releasing cytolytic granules in response to cit-vimentin or cit-H3, but not their native forms. Even though this cytotoxic activity occurred in an HLA-independent manner using an HLA-unmatched cancer cell line, this HLA-independent cytotoxicity was mediated by citrullinated antigen-activated ACPA+ RA CD8⁺ T cell lytic responses. Our results suggest that in ACPA+ RA, synovial CD8⁺ T cells could be activated by citrullinated antigens to mediate joint tissue destruction.

Further, it is possible that the HLA class I-independent killing activity of ACPA+ RA CD8⁺ T cells might be mediated by an expanded subpopulation of TCRγδ⁺CD8⁺ T cells, even though this subpopulation represents a low proportion of the CD8⁺ T cells in ACPA+ RA (< 5% of total CD8⁺ T cells as shown in Fig. 1e). It has been reported that TCRγδ⁺CD8⁺ T cells secrete pro-inflammatory and cytotoxic molecules like conventional TCRγδ⁺CD4⁻CD8⁻ T cells, and that this subpopulation could act as adaptive immune cells that are activated and mediate cytotoxic killing in response to specific antigens. Nevertheless, further investigation is needed to more fully determine if the TCRγδ⁺CD8⁺ T cells observed in ACPA+ RA are specific for and activated by citrullinated antigens. Thus, citrullinated proteins can induce ACPA+ RA CD8⁺ T cells to exert cytolytic activity which could contribute to synovitis and joint tissue destruction in RA.

In conclusion, we identified a previously unappreciated clonally expanded cytotoxic CD8⁺ T cells in the blood and synovium of ACPA+ RA patients. These cytotoxic CD8⁺ T cells are characterized by expression of pro-inflammatory and cytolytic mediators. We demonstrated that citrullinated antigens activate these cytotoxic CD8⁺ T cells in an HLA class I-dependent manner to expand, mediate effector activity, and kill target cells. Together, our findings suggest that cytotoxic CD8⁺ T cells target citrullinated antigens and could mediate

joint tissue destruction in RA, and provide rationale for development of therapeutics targeting cytotoxic CD8⁺ T cells.

## Methods
### Study design
The objective of this study was to investigate whether CD8⁺ T cells contribute to the pathogenesis of RA, including whether (i) CD8⁺ T cells target citrullinated antigens in RA and (ii) anti-citrullinated antigen CD8⁺ T cells in RA have the potential to mediate joint tissue destruction. We used flow cytometry and single-cell RNA sequencing to analyze the effector phenotypes and clonal expansion of CD8⁺ T cells in ACPA+ RA blood. To investigate the antigen targets of clonal lineage expanded CD8⁺ T cells in RA, we stimulated ACPA+ RA PBMCs with candidate citrullinated and native antigens. Blood samples were collected under institutional review board (IRB) approved protocols and after written informed consent at VA Palo Alto Health Care System and Stanford University (IRB3780). Clinical characteristics are provided in Supplementary Table 1 for all patients from which data were available. The blood samples of 12 RA patients were sequenced for HLA typing at Histocompatibility and Immunogenetics Laboratory in Stanford Blood Center (Supplementary Table 3). Healthy samples were randomly obtained through the Stanford Blood Center.

### Sample preparation
Whole blood was collected and PBMCs were isolated using Ficoll-Paque density gradient centrifugation (Sigma Aldrich, GE17-1440-03) for cellular profiling of CD8⁺ T cells. Cells were cryopreserved in Recovery Cell Culture Freezing Medium (Thermo Fisher Scientific, 12648010). For synovial tissue disaggregation, cryopreserved synovial tissues in Cryostor CS10 (Sigma-Aldrich, C2874-100ML) were obtained from hospital for special surgery. Each synovial tissue specimen was fully thawed at 37 °C and rinsed with RPMI 1640 (Corning Life Science, MT15040CV) containing 10% FBS (ATCC, 30-2020) and 1% glutamine (Gibco, 25030081). The tissue fragments were minced and digested with 100 μg/mL Liberase TL (Roche, 05401020001) and 100 μg/mL DNase I (Roche, 4716728001) in RPMI 1640 for 30 min at 37 °C with inverting to disaggregate the fragments into single cell suspensions. The digested fragments were filtered through a 70 μm cell strainer and washed with cold RPMI 1640 with 10% FBS and 1% glutamine before antibody staining and 10X single cell RNA sequencing.

### Flow cytometry and intracellular staining
Thawed PBMCs were stabilized at 37 °C overnight prior to staining with Fixable Viability Stain 510 (BD Bioscience, 564406), followed by fluorophore-conjugated antibodies targeting surface molecules in Stain Buffer (BD Bioscience, 554656) (Supplementary Table 4). For intracellular staining, PBMCs were re-stimulated with Cell Stimulation Cocktail (Thermo Fisher Scientific, 00-4975-93), fixed and permeabilized, then stained with intracellular molecule-targeting

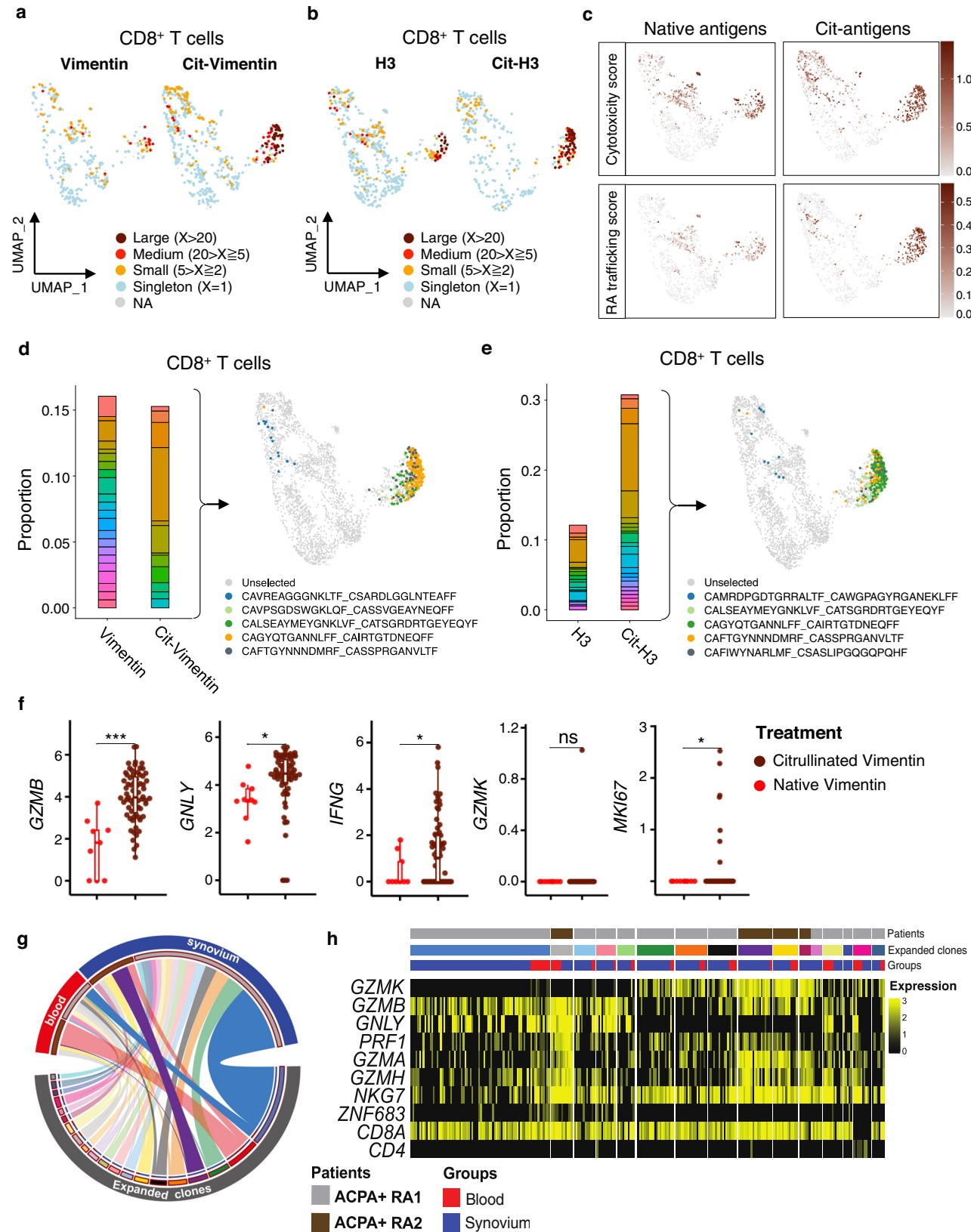

antibodies (Supplementary Table 4). To count absolute cell numbers of each population, Precision Count Beads (BioLegend, 424902) were added to the stained cells and samples were analyzed by flow cytometry. The absolute cell number was calculated based on the following formula: absolute cell count (cell/µl) = (cell count × Precision Count Beads volume)/(Precision Count Beads

count × cell volume) × Precision Count Bead concentration. The stained cells were analyzed using a BD LSR Fortessa or LSR II flow cytometer and the data collected by FACSDiva (BD Biosciences) and analyzed with FlowJo v.10.7.1 (Treestar). The antibodies used in flow cytometry analysis are listed in Supplementary Table 4 including fluorescence, dilution, company, and catalog number.

**Fig. 5 | Citrullinated antigens induce clonal expansion of ACPA+ RA blood CD8+ T cells that express cytotoxic and synovial-trafficking markers. a, b** UMAP plots showing clonal size of ACPA+ RA CD8+ T cells ($n = 3284$) stimulated by vimentin or citrullinated vimentin (**a**) and H3 or citrullinated H3 (**b**). **c** Cytotoxicity and RA synovial-trafficking scores in CD8+ T cells stimulated by native or citrullinated antigens. **d, e** Comparison of TCR repertoire in CD8+ T cells stimulated by native or citrullinated proteins (Vimentin; **d** or H3; **e**). Bar graphs representing changes of proportion in each top 5 clonotype between native and citrullinated antigens (*left*), UMAP depicting the selected 5 expanded clones labeled by different color (*right*). **f** Expression level of *GZMB*, *GNLY*, *IFNG*, *GZMK* or *MKI67* in the selected 5 clonotypes stimulated by native vimentin ($n = 9$) or citrullinated vimentin ($n = 67$). The boxplot represents the median shown as a line in the center of the box, the boundaries are the first and third quartile, and whiskers represent the minimum and maximum

values in the data. **g, h** Integrated analysis of paired blood and synovial tissue using single cell RNA seq datasets to identify whether RA synovial cytotoxic CD8+ T cells express transcriptional programs shared with those expressed by the citrullinated antigen-reactive clonally expanded *GZMB+GNLY+* CD8+ T cells in RA blood. Circos plot depicting shared expanded clonal families in PBMCs (red) or synovium (blue) from two representative ACPA+ RA patients distinguished by brown and gray colors (**g**). Heatmap showing expression level of cytotoxic markers such as *GZMB*, *GNLY* or *GZMK* in each expanded clone of blood or synovium (**h**). In **f**, $^{*}P < 0.05$, or $^{***}P < 0.001$ by two-tailed unpaired *t*-test (*GZMB*; $^{***}P = 0.00054$, *GNLY*; $^{*}P = 0.024$, *IFNG*; $^{*}P = 0.014$, *GZMK*; $P = 0.3209$, *MKI67*; $^{*}P = 0.016$). ns, not significant. Source data are provided as a Source Data file. ACPA anti-citrullinated protein antibodies, RA rheumatoid arthritis.

## Single-cell whole transcriptome and TCR sequencing

For single-cell RNA and TCR sequencing of CD8+ T cells, CD3+ T cells were isolated from PBMCs from RA and HC patients using the EasySep Human T cell isolation kit (StemCell Technologies, 17951) and analyzed using 10X Genomics protocols according to the manufacturer's instructions. Single-cell suspensions were loaded into a 10X chromium controller to capture -5000 cells per sample along with uniquely barcoded Gel beads. Single-cell libraries were prepared using Chromium Next GEM Single Cell V(D)J Reagent Kits v1.1. The barcoded cDNAs were pooled and sequenced using Illumina Novaseq 6000 and paired end reads. Using the cellranger mkfastq function (10X Genomics), base call files of for 5' gene expression libraries and V(D)J libraries were demultiplexed and converted into FASTQ files and mapped to the GRCh38 reference genome. TRA or TRB were annotated using the cellranger vdj function (10X Genomics).

## Computational analysis of single-cell RNA-seq data

Gene expression analyses of single cells was conducted using the R package Seurat (v.4.0.0)[46] to perform data scaling, transformation, clustering, dimensionality reduction, differential expression analyses and most visualization. The count matrix was filtered to remove cells with >10% of mitochondrial genes or low (<400) gene counts or less than 500 unique molecular identifier (UMI) gene counts. The UMI gene counts were scaled and normalized using the R package SCTransform[47]. The normalized data were integrated into single Seurat object, and input to the IntegrateData function to perform data integration.

Principal component analysis was performed using variable genes, and the first 30 principal components (PCs) were used to perform Uniform Manifold Approximation and Projection (UMAP) to embed the integrated dataset into two dimensions. Afterwards, the first 30 PCs were used to construct a shared nearest-neighbor graph (SNN), and this SNN used to cluster the cells. Cell annotation was performed using the R package SingleR (v.1.1.7) to determine cellular identity. The potential cell doublet was calculated using the scds R package[48] and clusters with high doublet score >0.8 were excluded.

CD8+ T cells were computationally identified based on the unsupervised clustering of the entire CD3+ T cells and defined as the CD8+ T cell clusters based on the gene expression and CITE-seq expression of CD8A measure by Seurat. From this clustering, we identified three clusters as CD8+ T cells. Afterwards, we selected the CD8+ T cells clusters and filtered out cells with greater than 1 of *CD4* level normalized mRNA expression. These resulted in 15,112 CD8+ T cells. The CD8+ T cells were re-clustered with newly calculated principal components. Thereafter, the R package Harmony[49] was used to normalize across batches. Differentially expressed genes within each cluster were analyzed to display the heatmap with the top 10 genes in each cluster using FindMarkers function with Wilcoxon Rank Sum test and the

Benjamini−Hochberg method to adjust the *P* values for multiple testing. CD8+ T cell were annotated using the expression level of *GZMB*, *GZMK* or canonical naïve/memory markers.

CITE-seq antibody counts were normalized using the centered log ratio transformed (CLR) counts. The used CITE-seq antibodies are listed in Supplementary Table 5.

## Pseudobulk analysis of differentially expressed genes and pathways in scRNA data

To identify differentially expressed genes (DEGs) in *GZMB+* or *GZMK+* clusters between ACPA+ RA and HC, we performed differential expression testing using the pseudobulk RNA expression.

We computed the pseudobulk gene expression for each cluster of each sample using the scater R package[50]. Samples with less than 10 cells present in a cluster were excluded from the pseudobulk expression calculation of that cluster. Thereafter, the pseudobulk expression counts were normalized using DESeq2 R package[51]. The normalized pseudobulk expression counts were used for differential gene expression analysis of ACPA+ RA vs. HC using DESeq2 R package. The calculated log2 fold change of gene expression in ACPA+ RA vs. HC was multiplied by the -log (FDR, false discovery rate) and used to rank the genes descending and perform gene-set enrichment analysis (GSEA) for *GZMB+* cluster and *GZMK+* cluster. The normalized enrichment score (NES) and adjusted *P* value for multiple testing were used to define the enriched biological process pathways in either *GZMB+* or *GZMK+* clusters. The gene sets of gene ontology were obtained from Molecular Signatures Database (v.7.5.1) and the GSEA was performed using the fgsea R package[52].

## Analysis of TCR clonotypes paired with scRNA data

The TCRs were annotated using the cellranger vdj pipeline. The scRepertoire R package v.1.1[53] was used to identify and analyze TCR clonotypes based on TCR alpha and beta chains and CDR3 sequences. The clonotype data was annotated using Seurat to generate gene expression and cluster information using the combineExpression function. Cells with the same paired TCRαβ genes were annotated as expanded clonal lineages. The chord diagram to visualize the shared clones between each cluster was generated by getCirclize function using the R package circlize[54], and the heatmap was displayed with the R package ComplexHeatmap v.2.13.1[53,55]. The shared expanded TCR clonal families in paired blood and synovial samples were visualized using circlize R package.

Here is the version of each R package we used.

ComplexHeatmap_2.12.1, circlize_0.4.15, fgsea_1.22.0, DESeq2_1.36.0, scater_1.24.0, ggplot2_3.3.6, scuttle_1.6.3, SingleCellExperiment_1.18.0, SummarizedExperiment_1.26.1, Biobase_2.56.0, GenomicRanges_1.48.0, GenomeInfoDb_1.32.4, IRanges_2.30.1, S4Vectors_0.34.0, BiocGenerics_0.42.0, MatrixGenerics_1.8.1, matrixStats_0.62.0, sctransform_0.3.4, harmony_0.1.0, Rcpp_1.0.9, scds_1.12.0.

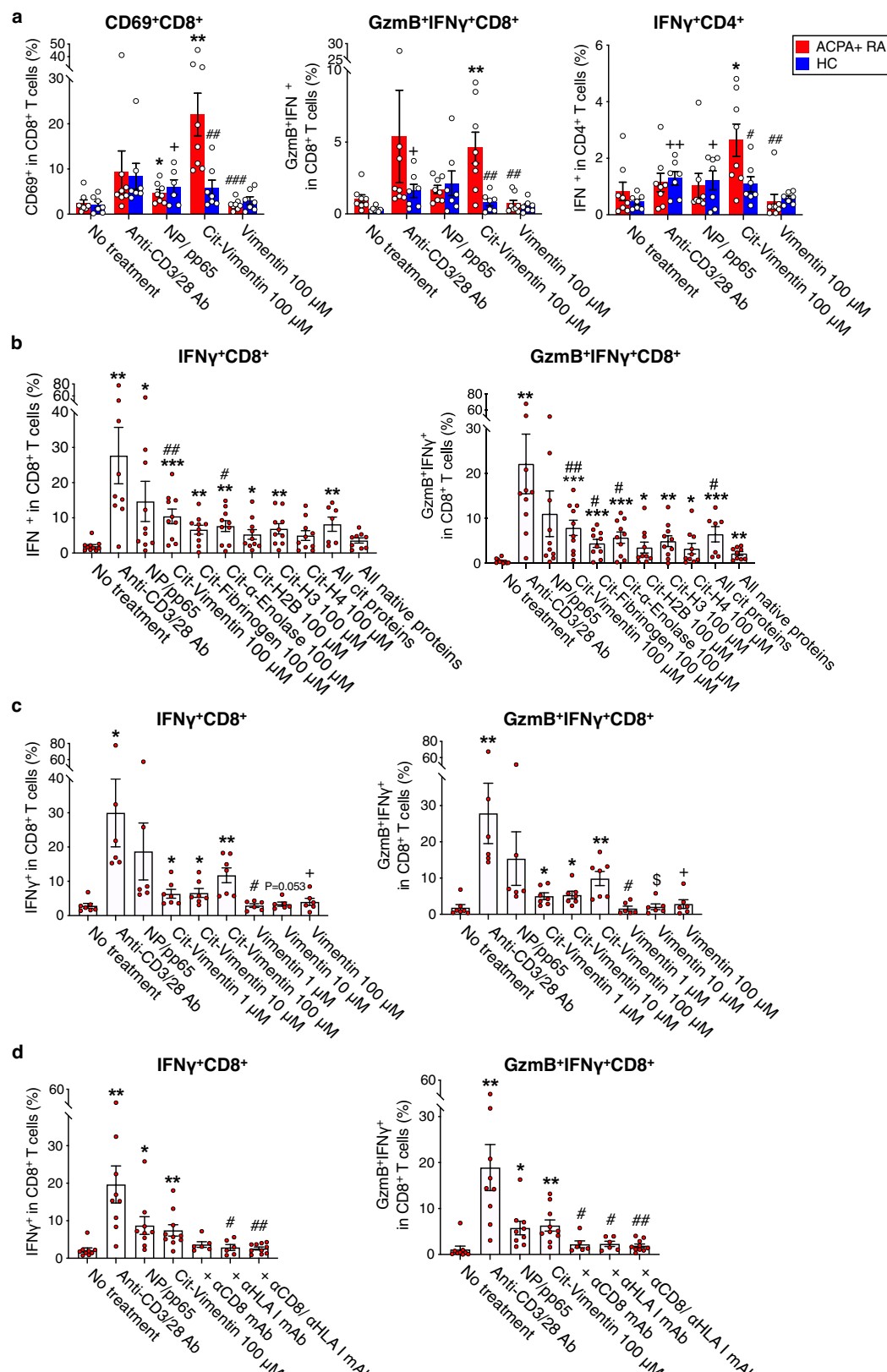

## Citrullinated protein in vitro stimulation of fresh whole blood or PBMCs

Within 4 h of blood collection, 1 ml of whole blood was incubated with 2 μg/ml anti-CD28 (BD Biosciences, 555725) and CD49d (BioLegend, 304302) antibodies, 50 μM each of recombinant influenza A H1N1 nucleoprotein (NP, Sino Biological, 11675-V08B) and CMV pp65

(Miltenyi Biotec, 130-091-824) protein, 100 μM of cit-vimentin (Cayman Chemical, 21942), 100 μM native vimentin (Cayman Chemical, 11234), or 1 μg/ml anti-CD3 antibodies (BD Biosciences, 566685) were added, followed by incubation at 37 °C for 6 h. In the last 5 h, Golgi inhibitor Brefeldin A (10 μg/ml) was added, and then addition of 20 mM EDTA for 15 min followed by FACS lysing solution (BD Biosciences, 349202) to

**Fig. 6 | ACPA+ RA blood CD8+ T cells upregulate activation and cytotoxic markers in response to citrullinated antigens. a** Stimulation of fresh blood from ACPA+ RA ($n = 8$, red bars) or HCs ($n = 7$, blue bars) with anti-CD3/28 Abs, NP/pp65, cit- or native vimentin for 6 h with addition of Golgi inhibitor for the last 5 h. Quantification of CD69+ and GzmB+IFNγ+ CD8+ T cells, and IFNγ+ CD4+ T cells. For CD69+CD8+, *$P = 0.049$, *$P = 0.0397$, **$P = 0.001$, ##$P = 0.009$, or ###$P = 0.0008$. For GzmB+IFNγ+CD8+, +$P = 0.0175$, **$P = 0.0071$, ##$P = 0.0071$, or ##$P = 0.0037$. For IFNγ +CD4+, ++$P = 0.0065$, *$P = 0.0489$, *$P = 0.0155$, #$P = 0.0351$, or ##$P = 0.0038$. **b** PBMCs of ACPA+ RA patients ($n = 10$) were incubated with anti-CD3/28 Abs, NP/pp65 (50 µM of each), individual citrullinated proteins (100 µM), all citrullinated proteins (20 µM each), or all native proteins (20 µM each) for 16 h. Percentage of IFNγ or Granzyme B (GzmB) expressing CD8+ T cells measured by intracellular staining. **c** Quantification of IFNγ+ or GzmB+IFNγ+ expressing CD8+ T cells in ACPA+ RA PBMCs treated with cit- or native vimentin in a dose-dependent manner (1, 10 or 100 µM) ($n = 7$). **d** Quantitative analysis of IFNγ+ or GzmB+IFNγ+ expressing CD8+

T cells in ACPA+ RA PBMCs stimulated with cit-vimentin in the presence or absence of anti-CD8 and/or HLA class I-blocking antibody ($n = 10$). Data are presented as means ± SEM. For **a**, two-tailed unpaired $t$-test: *$P < 0.05$ or **$P < 0.01$ versus no treatment in ACPA+ RA; +$P < 0.05$, or ++$P < 0.01$ versus no treatment in HC; and #$P < 0.05$, ##$P < 0.01$, or ###$P < 0.001$ versus cit-vimentin in ACPA+ RA. For **b**, two-tailed unpaired $t$-test: *$P < 0.05$, **$P < 0.01$, or ***$P < 0.001$ versus no treatment; #$P < 0.05$, or ##$P < 0.01$ versus all native proteins. For **c**, two-tailed unpaired $t$-test: *$P < 0.05$, or **$P < 0.01$ versus no treatment; #$P < 0.05$ for cit-vimentin 1 µM versus vimentin 1 µM; $$P < 0.05$ for cit-vimentin 10 µM versus native vimentin 10 µM by two-tailed unpaired $t$-test; +$P < 0.05$ for cit-vimentin 100 µM versus native vimentin 100 µM. For **d**, by two-tailed unpaired $t$-test: *$P < 0.05$, or **$P < 0.01$ versus no treatment; #$P < 0.05$, or ##$P < 0.01$ versus cit-vimentin 100 µM. Source data are provided as a Source Data file. HC healthy control, ACPA anti-citrullinated protein antibodies, RA rheumatoid arthritis, Cit-Vim citrullinated vimentin.

lyse RBCs. After the centrifugation, cells were stained with fluorophore-conjugated anti-CD4, CD8 or CD69 for surface staining and anti-granzyme B (GzmB) or anti-IFNγ antibodies for intracellular staining.

For the PBMC stimulation, cryopreserved PBMCs were thawed, stabilized in RPMI 1640 with 5% human serum (Sigma Aldrich, H3667-100ML) overnight at 37 °C, and seeded into 96 round-bottom plates. Cells were pre-incubated with 10 µg/mL Polymyxin B (Sigma-Aldrich, 92283) for 30 min and then incubated with the recombinant proteins or anti-CD3 antibody (1 µg/ml) for 16 h. 2 µg/ml of anti-CD28 and anti-CD49d antibodies were added to each well. The human recombinant proteins included NP, pp65; citrullinated proteins including cit-vimentin, cit-fibrinogen, cit-α-enolase, and histones cit-H2B, cit-H3 and cit-H4 (all from Cayman Chemical, 400076, 21585, 30133, 17926, and 17927); and the native forms of vimentin, fibrinogen, α-enolase, and histones H2B, H3.1 and H4 (Cayman Chemical; 25151, Thermo Fisher; RP43142, New England Biolabs; M2505S, M2503S, M2504S). To block the interaction of HLA class complex and CD8+ T cells, purified anti-human CD8 and/or HLA-A, B, C antibody (1 µg/ml, BioLegend; 344702, 311402) was added concomitantly with cit-vimentin or cit-H3. In the last 5 h, eBioscience Protein Transport Inhibitor Cocktail (500X, Thermo Fisher, 00-4980-93) was added to detect intracellular staining. After 16 h, the cells were intracellularly stained with fluorophore-conjugated anti-CD4, CD8, GzmB and IFNγ antibodies and detected with BD LSR Fortessa flow cytometer.

### T cell proliferation assay

For the isolation and maturation of monocyte-derived dendritic cells, PBMCs were seeded in 24 well plates, and after 1-day adherent monocytes cultured in Differentiation media containing Serum-Free Dendritic Cell Base Media with IL-4 and GM-CSF (R&D Systems, CCM003) for 5 days to generate immature dendritic cells. On day 5, fresh Differentiation media with recombinant human TNF was added to induce dendritic cell maturation for an additional 2 days. Mature dendritic cells were incubated with 50 µM each of recombinant NP and pp65 protein, or 100 µM of cit-vimentin or native vimentin for 6 h. The antigen-loaded DCs were cocultured with T cells. For the labeling of CD3+ or CD8+ T cells, CD3+ or CD8+ T cells were isolated using EasySep Human T cell isolation kit or EasySep Human CD8+ T Cell Isolation Kit (StemCell Technologies, 17953) and labeled with eBioscience™ Cell Proliferation Dye eFluor 450 (Thermo Fisher, 65-0842-85). For the coculture, the antigen-loaded DCs were cocultured with the labeled CD3+ or CD8+ T cells at 1:10 ratio in the culture media with recombinant IL-2 (50 IU/ml, Peprotech, 200-02), IL-7 (10 ng/ml, Peprotech, 200-07), and anti-CD28 and CD49d antibodies (2 µg/ml) at 37 °C for 10 days. To inhibit the HLA class I recognition by CD8+ T cells, anti-CD8 and/or HLA-A, B, C antibody (1 µg/ml) was added with cit-vimentin. Culture media was replenished every 3 days. After 10 days, cells were stained with fluorophore-conjugated anti-CD4 and anti-CD8 antibodies, and proliferative capacity measured by flow cytometry. The proliferating

cells were quantified based on the proportion of cells in Cell Proliferation Dye$^{low}$ population.

Single cell RNA sequencing data of citrullinated antigen-stimulated T cells was analyzed using the same workflow with the previous methods. The scores of cytotoxicity, RA synovium trafficking, proliferation and inflammation were quantified by the gene expression levels of each gene marker listed in Supplementary Table 6.

### CD107a degranulation assay

For the detection of the degranulation process in CD8+ T cells, PBMCs were incubated with anti-CD3 antibody (1 µg/ml) or each protein. In the last 5 h, fluorophore-conjugated anti-CD107a antibody and eBioscience Protein Transport Inhibitor Cocktail (500X) was added to detect CD107a levels.

### LDH cytotoxicity assay

The cytotoxic capacity of CD8+ T cells was assessed by measuring Lactate dehydrogenase (LDH) in the CD8+ T cell and tumor cell coculture supernatants. $10^4$ DLD-1 colorectal adenocarcinoma (ATCC, ATCC CCL-221) cells were seeded in 96 flat-bottom well plates and cocultured with isolated CD8+ T cells at the corresponding E:T ratio (1:1, 5:1 or 10:1) using triplicate wells. Using the Cytotoxicity Detection Kit$^{PLUS}$ (Sigma Aldrich, 4744926001), supernatant LDH was measured. The LDH levels in DLD-1 tumor cell alone cultures were used as a low control, and the lysis solution as a high control. The percentage of cytotoxicity = (experimental value − low control)/(high control − low control) × 100 (%).

### Statistical analysis

Data are represented as means ± SEM. Two-tailed unpaired student's $t$ tests were used to compare between two groups, and ordinary one-way or two-way analysis of ANOVA with Tukey's multiple comparisons test was used when multiple groups were compared. Spearman's correlation was used to analyze correlations between two variables. GraphPad Prism v9 was used for the analyses.

### Reporting summary

Further information on research design is available in the Nature Portfolio Reporting Summary linked to this article.

### Data availability

The raw single cell RNA gene expression, V(D)J and CITE-seq data generated in this study have been deposited in the Sequence Read Archive (SRA) database under accession code PRJNA900189. The differentially expressed gene data, CITE-seq list, or gene list used to measure scores of cytotoxic, proliferation or RA trafficking generated in this study are provided in the Supplementary Information. All other data are available in the article and its Supplementary files or from the

corresponding author upon reasonable request. Source data are provided with this paper.

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

## Acknowledgements

We thank R. Camille Brewer, Tobias Lanz and Alejandro Gomez for insightful discussions. We also thank healthy donors at Stanford Blood Center. The study was supported by following grants: National Institutes of Health grant RO1 ARO63676 (WHR), National Institutes of Health grant U19 AI110491 (WHR), funding from Janssen Biotech, Inc. (WHR), National Institutes of Health grant RO1 ARO78268 (WHR), National Institutes of Health grant U01 AI101981 (VMH, KDD, EAJ, JHB, WHR), and Knut and Alice Wallenberg Foundation (SY).

## Author contributions

J.S.M., S.Y. and W.H.R. conceptualized the study and designed the experiments. O.S. and K.S. collected RA patient blood samples and evaluated clinical data of each patient from VA Palo Alto healthcare system. S.G. and L.D. collected and processed RA paired blood and synovium from HSS FLARE cohort. J.S.M., S.Y., N.S.R., N.L.R., S.B., J.C., E.A.J., J.H.B., K.D.D., V.M.H. and M.M.D. contributed to the review of this study. J.S.M., S.Y., N.S.R. and R.I. performed 10X library preparation. S.Y. processed and analyzed single-cell RNA seq data. S.Y. performed and visualized computational analyses. J.S.M. analyzed and visualized the in vitro experiments. W.H.R. and V.H.M. acquired the funding to support this study. J.S.M., S.Y., M.M.D., E.A.J., J.H.B., K.D.D., V.M.H. and W.H.R. wrote and edited the manuscript.

## Competing interests

The authors declare no competing interests.
