## [Peer Review File · Nature Communications]

Cytotoxic CD8+ T cells target citrullinated antigens in rheumatoid arthritisREVIEWER COMMENTS

Reviewer #1 (Remarks to the Author):

This is a nice study that combines single cell transcriptomics with functional assays to demonstrate that peripheral blood of ACPA+ RA patients harbors cytotoxic CD8+ T cells that specifically respond to citrullinated protein antigens in an HLA Class I-dependent manner by clonally expanding and expressing a cytotoxic program. This pulls together a few disparate threads related to the role of CD8+ T cells in RA, adds a new dimension to our understanding of pathogenic mechanisms in rheumatoid arthritis and opens up new avenues for understanding, monitoring and treating this autoimmune disorder.

The Discussion makes claims about the synovium that are too strong, given that the data in this study are from peripheral blood. Either the authors should provide more convincing analyses of published CD8+ T cell data from synovial samples or they should soften this aspect of their conclusions.

Also, the single cell data are limited in size and analyzed in a manner that ignores sample-specific artifacts. Consequently, many of the granular conclusions from single cell analysis may either be incorrect or only weakly supported, as described below. However, this is fixable. The single cell data analysis problems can (and should!) be addressed and the unnecessary and unsupported conclusions should be discarded. What's important is that the major biological conclusions appear to be both valid and important for the field.

Major comments

There are quite a few problems with the clustering analysis of scRNA-seq data. The distinctions between scRNA-seq clusters in Fig. 2a and Extended Data Fig. 3c are very subtle. It's not clear that these are biologically distinct cell subtypes. For example, in the heatmap in Extended Data Fig. 3c, there's no clear difference between the GZMB+ GNLY+ ZNF683+ cell type and the GZMB+ GNLY+ cell type. These two clusters could merely represent technical variation within the same cell type. Or perhaps batch effects. Or arbitrary partitioning of cells by a clustering algorithm that has been asked to generate more clusters than the data can justify. Or a sample-specific bias (see below). Sample id should be indicated for each cell.

Similarly, the naming of the "GZMB-low GNLY+" cluster is somewhat arbitrary, because GZMB is not the only marker that's low in this group of cells. In fact, virtually every marker gene has lower expression in this cluster than in the GZMB+ GNLY+ cluster, which gives the impression that the expression difference between two clusters may be technical (data quality) rather than biological. In general, biological differences manifest as strong changes in the expression of a relatively small number of genes. On the other hand, technical differences drive mild expression changes in a large number of genes.

Naïve (i) and Naïve (ii) clusters look more or less identical in Extended Data Fig. 3c, which reinforces the concern that the data have been overclustered.

Could the authors merge some of these highly similar scRNA-seq clusters and still support their major conclusions? It's common in the single cell field to manually collapse neighbouring clusters based on marker expression, particularly if there are no convincing DE genes between them.

Alternatively, the authors could use a lower setting of the resolution parameter in Seurat – that would also reduce the number of clusters.

To test for technical variation in scRNA-seq as a confounding factor in clustering, it would be useful to color the CD8+ umap plot by QC metrics: number of features and proportion of mitochondrial reads. For this plot to be useful, one would need to use a color scale that showed some contrast between the 20th percentile and the 80th percentile of the QC metric.

To test for batch effects, the same umap should be colored by patient, taking care to show the umap in a large enough panel, with a color scheme that shows some contrast between patients. The main “advantage” of the default Seurat color scheme is that it gives you colors that are indistinguishable from one another. This makes the figures uninterpretable and therefore irreproachable. You can make any claim you want and the reviewer or reader has no way of disproving it because they can't tell the colors apart. This has no doubt contributed to the huge popularity of the Seurat software package. But it's not good for science. A lot of bogus claims have slipped unchallenged into the single cell literature because of this uninformative color scheme. I would recommend a different set of colors.

Why do the proliferating CD8 T cells look like they express markers of all subtypes (Extended Data Fig. 3c again)? Are these cells merely doublets? To annotate them as proliferative, one must show that they preferentially express proliferation markers.

Another problem with the clustering is that naïve-like and effector-like cells are present within the same cluster (for example, see the Memory-like Naïve and Proliferating clusters in Extended Data Fig. 3c). Again, this suggests that cells are clustering by technical variation or batch rather than by biology, at least to some extent.

Figure 2b, c: only the GZMB-low GNLY+ cluster shows a significant difference between HCA and RA – the other cell types cited in the text are not convincingly different. Unfortunately, GZMB-low GNLY+ is the slightly dubious cell subtype with no clear distinguishing markers (see above).

Figure 2d and f are based on the default DE gene analysis module in the Seurat package, which is not statistically valid because it assumes that each cell is a statistically independent replicate. There is a growing body of literature on this artifact and how to address it (see Nat Commun 2021; 12:738 and references therein). Again, this is a case where Seurat is popular because it gives you the result you want (in this case: spectacular DE gene p-values). In reality, the transcriptomes of cells within a single sample are highly correlated, so these p-values are misleading. A more statistically valid approach would be to construct the pseudobulk transcriptome of each sample and then perform DE analysis using DESeq or EdgeR (<https://www.biorxiv.org/content/10.1101/2022.02.16.480517v1>). This will give you very few DE genes if your cohort size is small. But they will be credible.

Also, it's not appropriate to use a raw p-value cutoff of 0.05 in DE gene analysis. There has to be some correction for multiple testing, since thousands of genes are being tested.

From what I could tell, the GO term enrichment analysis in Fig. 2e, g compares DE genes to all genes in the genome. This is not a valid approach. DE genes need to be compared to the set of all genes expressed in CD8+ T cells, since a gene needs to be expressed to have a chance of being differentially expressed.

Figure 3e indicates that the transcriptomes of cells from the same sample are indeed highly correlated with each other. For example, most of the RA cells expressing the gamma-delta T cell marker TRGV2 come from just 1 of the 9 RA samples. Most of the panels in Figures 2 and 3 ignore this profound sample-specificity. Based on the data analysis strategy used in these two figures, one would conclude that TRGV2 was uniformly upregulated in RA, but in fact it's only detectably upregulated in one single RA sample. Similarly, by ignoring sample-to-sample variation, the authors might conclude from Figure 3e that TRGV2 was a marker of all clonally expanded CD8+ T cells. But again, it's only a marker of clonally expanded cells from one RA sample. Many key markers discussed in this manuscript, including GZMB, GNLY, TBX21 and ZNF683, show similarly strong sample specificity in Figure 3e, which deepens the concern regarding the validity of naively aggregating all cells from all samples. To address this concern, most of the panels in these two figures need to be redone. The authors should confirm that their conclusions reflect trends that are consistent across samples rather than driven by one or two outliers. This would involve pseudobulk analysis, examination of the consistency of trends across RA samples, etc.

Another concern after examining Figure 3e is that 80-90% of RA cells come from just 4/9 RA samples. Since the analyses here are mostly based on averaging across all cells, this implies that most of single cell conclusions are essentially derived from only 4 RA samples.

Extended Data Fig. 5d reveals major technical confounders in the scRNA-seq data. 8-9 genes at the bottom of this heatmap are labeled as being associated with memory function, but the three genes at the bottom of this set (IL7R, FOS and JUN) form a coregulated module that is anti-correlated with the rest of the memory markers and in fact anti-correlated with most of the marker genes in this panel. These genes most likely represent stress response, rather than memory T cell function. Almost all cell clusters contain a subset of cells that strongly upregulate these three genes in unison. These three genes are often upregulated in response to the stresses cells face in the course of isolation and processing for scRNA-seq. I would suggest that the stressed cells be removed from the analysis, or at least analyzed separately.

Even if one ignores the putative stress-responsive cells, the Central Memory and Memory-like Naïve clusters in Extended Data Fig. 5d show profound intra-cluster expression variation that's larger than the difference between these two clusters. In either cluster, one cell subgroup expresses naïve and memory markers, while another expresses cytotoxic effector markers. The same can be said of the "proliferating" cluster. Thus, a substantial fraction of the cells labeled as memory or memory-like or naïve may actually be effector cells.

More technical artifacts: in Extended Data Figure 5d, The "Naïve (i)" and "GZMG+ GNLY+ ZNF683+" clusters show a blurry vertical stripe, i.e. a patch of cells that don't show preferential expression of any particular marker. These are most likely low-quality cells, potentially derived from 1-2 low-quality samples. Perhaps those samples should be discarded.

Having read a bit further, I would say that my concerns regarding sample-specific effects vs trends shared across samples applies not just to Figures 2 and 3 but to most or all of the main and supplementary figures that are based on scRNA-seq data.

Despite all the data quality and data analysis concerns, the conclusion that "the large clonal expansions of CD8+ T cells are predominantly in the GZMB+ clusters as compared to GZMK+ clusters" does appear to be true. However, the word "predominantly" may be a bit too strong.

Another major conclusion, namely that "CD8+ T cells in ACPA+ RA blood exhibit proliferative responses to cit-vimentin in a HLA class I-dependent manner," also appears to be supported by the data (Figure 4).

Figure 5: the conclusions are again very plausible, but again no attempt is made to determine if they are driven by one single atypical sample or if they represent a shared trend across all samples (cells are only analyzed in aggregate across all samples).

It's not clear what one could conclude from Figure 5g, h. These panels show that there are cells in the synovium that are transcriptomically similar to the GZMB and GZMK-expressing populations in peripheral blood. But the degree of similarity is not obvious. Also, these results don't tell us anything about whether or not the GZMB-expressing synovial cells are clonally expanded. What is the message here? Do we need these two panels?

The responsiveness of RA CD8+ T cells to citrullinated vs native vimentin also appears to be a robust conclusion (Figure 6).

Minor comments

scRNA data: how many single cells were analyzed from each sample? Were there any low-quality samples that should have been discarded? Or any samples that contributed a negligible number of cells, too few to discern any meaningful trend? I suspect the answer is yes to the latter question.

The term CCP is first used on pg. 4 but defined only on pg. 12.

“CD8A >0.5 and CD4 <0.1” – in what units are the expression levels quantified?

Extended Data Fig. 3: what does “fold change >0.8” mean? Is this a log-fc? Log2? Log10? There are multiple places in the manuscript where numbers are given without explaining the units or how they were calculated.

Extended Data Fig. 3c: please use the same cluster colors as in Fig. 2a.

The Methods section states that the authors initially clustered all cells and then, in a second round, clustered only CD8+ T cells. I could only find figures and results from the latter. Was the all-cell clustering result ever used in the manuscript? And if it was not used, why was it done?

CD8+ T cells in scRNA-seq data were defined based on these FACS-like criteria: CD3E >0.5 & CD8A >0.5 & CD4 <0.1. This is not a reliable way of analyzing scRNA-seq data, which have higher noise and dropouts than FACS data. The more conventional way to identify cell types is to cluster the single cell transcriptomes and map each cluster (or group of clusters) to a cell type or subtype based on markers. CD8+ T cells should be identifiable in this manner.

How was “batch” defined in the batch correction step? Was each sample a batch?

Figure 2c: the p-value is apparently obtained from two-way ANOVA. Which two factors were considered in this two-way analysis? The bar graph should be replaced by a figure showing the actual distribution of data points. There are only 14 data points, so it should be possible to show all of them as dots. This way, we can determine if the ANOVA results are credible or if they are driven by one or two outlier data points (ANOVA assumes a normal distribution, so one or two outliers can trigger a strong p-value).

Similarly, Figure 2h should show the expression level in each of the 14 samples and the De gene p-value should be specified for each gene, so we can tell if the expression difference is significant.

GNLY is one of the major markers used to label CD8+ cell subtypes - it should be shown in Figure 2b.

The umaps in Figures 2a and 3a should be the same(?).

Figure 3e is more useful than some of the other heatmaps, because disease status and sample ID are indicated in the top bar. It would be useful to show these covariates in the other heatmaps as well. Plus QC metrics such as mitochondrial read fraction and number of features.

Reviewer #2 (Remarks to the Author):

The paper by Moon et al investigate the presence and function of CD8+ T cells in Rheumatoid Arthritis and the correlation with ACPA positive disease. The study focusses on T cells from blood and define a subpopulation of GZMK/CD8+ population. Further they study the activation by citrullinated antigens. I believe the study is interesting but have some concerns:

Major issues:

1. The study also includes data on $\gamma\delta$ CD8+ cells that are thought to not be MHC restricted at least not to recognize peptides like other CD8+ T cells. What is their role here and since they are expanded are they affecting the data with killing that in part is “MHC independent”? I find this data curious, and it should be discussed both in the introduction and discussion part. Here they are lumped with CD8+ cells in general which might not be adequate.

2. It has been known for some time that ACPA positivity correlate with CD8+ T cells numbers (for example: de Hair 2013). This data needs to be referenced and other mechanisms discussed, such as the study by Jung et al that showed recently that CD8 T cells are present in the synovium and their activation is coupled with citrullination of antigen. They also show TCR repertoire and skewing of this. To me this is more reasonable as Antibodies are coupled to CD4 T cell activation which typically balance a CD8 response. Thus, one would expect more CD8 T cell activation in the ACPA- group of patients.

3 Much of the data is shown as proportions or percentages. For some of the key data it would be nice with absolute numbers.

4. In figure 1F the data for ACPA- RA should be shown.

5. In Figure 4 the proliferation to citrullinated antigen needs a control. Either HC or ACPA- RA or both.

Minor issues,

-The manuscript needs to be edited for clarity. Some sentences are very long and there is some run on

Reviewer #3 (Remarks to the Author):

This is an interesting study where the authors have investigated the CD8 T cell response to citrullinated self antigens in rheumatoid arthritis where CD4 T cells are more commonly associated. The study has been well performed and includes important controls such as ACPA negative patients. The results demonstrate that CD8 T cells are clonal expanded and activated in RA, have relevant cytotoxic and joint homing markers as well as the capacity to kill target cells in an in vitro assay. The authors have been careful and circumspect in their language as they are unable to demonstrate a direct causative link between CD8 T cells, their activation and pathology in RA patients. Overall, the work highlights the potential contribution of these cells and therefore is a relevant contribution. It would perhaps be important to examine a number of additional things to demonstrate cause and effect. 1. Responses to irrelevant citrullinated peptides. 2. In vitro and/or in vivo demonstration of pathological effects of CD8 T cells on joint cells using e.g. organoids, tissues slice or humanised murine models.

Reviewer #1 (Remarks to the Author):

This is a nice study that combines single cell transcriptomics with functional assays to demonstrate that peripheral blood of ACPA+ RA patients harbors cytotoxic CD8+ T cells that specifically respond to citrullinated protein antigens in an HLA Class I-dependent manner by clonally expanding and expressing a cytotoxic program. This pulls together a few disparate threads related to the role of CD8+ T cells in RA, adds a new dimension to our understanding of pathogenic mechanisms in rheumatoid arthritis and opens up new avenues for understanding, monitoring and treating this autoimmune disorder.

The Discussion makes claims about the synovium that are too strong, given that the data in this study are from peripheral blood. Either the authors should provide more convincing analyses of published CD8+ T cell data from synovial samples or they should soften this aspect of their conclusions.

Also, the single cell data are limited in size and analyzed in a manner that ignores sample-specific artifacts. Consequently, many of the granular conclusions from single cell analysis may either be incorrect or only weakly supported, as described below. However, this is fixable. The single cell data analysis problems can (and should!) be addressed and the unnecessary and unsupported conclusions should be discarded. What's important is that the major biological conclusions appear to be both valid and important for the field.

Major comments

1.1 *There are quite a few problems with the clustering analysis of scRNA-seq data. The distinctions between scRNA-seq clusters in Fig. 2a and Extended Data Fig. 3c are very subtle. It's not clear that these are biologically distinct cell subtypes. For example, in the heatmap in Extended Data Fig. 3c, there's no clear difference between the GZMB+ GNLY+ ZNF683+ cell type and the GZMB+ GNLY+ cell type. These two clusters could merely represent technical variation within the same cell type. Or perhaps batch effects. Or arbitrary partitioning of cells by a clustering algorithm that has been asked to generate more clusters than the data can justify. Or a sample-specific bias (see below). Sample id should be indicated for each cell.*

Response 1.1

We appreciate the Reviewer highlighting these points. For the RA and healthy patient samples analyzed, we added additional RA and healthy comparator samples and removed 2 samples from the previous dataset due to low cell numbers. The updated dataset now includes 6 healthy controls and 12 ACPA+ RA patient samples analyzed by scRNA-seq. We performed QC analysis based on generally accepted parameters, including that the data from each sample contain: (i) <10% percent mitochondrial genes; (ii) > 500 total distinct mRNAs per cell [based on UMIs]; (iii) > 400 distinct genes per cell; and (iv) < 0.8 predicted cell doublets [see our response to Reviewer 1 point 1.30, below]. We also now cluster the cells using moderate resolution that results in 7 major clusters of CD8+ T cells, and we believe our new clustering thresholds avoid over-clustering. The new clusters include naïve, GZMB+ GNLY+, GZMB+ KIR+, TCRgd+, GZMK+, memory, and CCR6+ CD161+ populations of CD8+ T cells. Based on the updated analysis, we identified one GZMB+ GNLY+ cluster, which is distinct from the other clusters. We also now provide a heatmap of marker expression for each cluster that includes sample IDs, to demonstrate to readers that there is no substantial sample-specific bias (Extended Data Fig. 4c).

We revised the manuscript to include our refined results, and updated Fig. 2, Fig. 3 and Extended Data Fig. 4.

Extended Data Fig. 4c (new):

1.2 Similarly, the naming of the “GZMB-low GNLY+” cluster is somewhat arbitrary, because GZMB is not the only marker that’s low in this group of cells. In fact, virtually every marker gene has lower expression in this cluster than in the GZMB+ GNLY+ cluster, which gives the impression that the expression difference between two clusters may be technical (data quality) rather than biological. In general, biological differences manifest as strong changes in the expression of a relatively small number of genes. On the other hand, technical differences drive mild expression changes in a large number of genes.

Response 1.2

We appreciate the Reviewer’s point, and now address this important point using expanded and more highly QC-controlled datasets as described in Response 1.1. Based on our expanded and refined datasets, we re-clustered the CD8+ T cells and now only have one GZMB+ GNLY+ cluster (revised Fig. 2a).

Revised Fig. 2a:

1.3 Naïve (i) and Naïve (ii) clusters look more or less identical in Extended Data Fig. 3c, which reinforces the concern that the data have been overclustered.

Response 1.3

We appreciate the Reviewer's point, and address this in our revised manuscript with our expanded and further QC'd datasets as detailed in Response 1.1. To avoid over-clustering, we now use more modest thresholds for clustering that result in only one Naïve cluster in the current data (revised Fig. 2a).

1.4 Could the authors merge some of these highly similar scRNA-seq clusters and still support their major conclusions? It's common in the single cell field to manually collapse neighbouring clusters based on marker expression, particularly if there are no convincing DE genes between them.

1.5 Alternatively, the authors could use a lower setting of the resolution parameter in Seurat – that would also reduce the number of clusters.

Response 1.4 and 1.5

As described in our Responses to 1.1, 1.2, and 1.3, we have now removed samples with low cell number samples, added additional samples, and refined our clustering to use more modest clustering thresholds. Based these refinements, highly similar naïve CD8⁺, memory CD8⁺, and GZMB⁺ CD8⁺ clusters have now been merged (revised Fig. 2a).

1.6 To test for technical variation in scRNA-seq as a confounding factor in clustering, it would be useful to color the CD8⁺ umap plot by QC metrics: number of features and proportion of mitochondrial reads. For this plot to be useful, one would need to use a color scale that showed some contrast between the 20th percentile and the 80th percentile of the QC metric.

Response 1.6

We appreciate the Reviewer's suggestion, and now provide the QC metrics to show that the clusters are not biased by the mitochondria percentage or low read counts (see directly below).

1.7 To test for batch effects, the same umap should be colored by patient, taking care to show the umap in a large enough panel, with a color scheme that shows some contrast between patients. The main “advantage” of the default Seurat color scheme is that it gives you colors that are indistinguishable from one another. This makes the figures uninterpretable and therefore irreproachable. You can make any claim you want and the reviewer or reader has no way of disproving it because they can’t tell the colors apart. This has no doubt contributed to the huge popularity of the Seurat software package. But it’s not good for science. A lot of bogus claims have slipped unchallenged into in the single cell literature because of this uninformative color scheme I would recommend a different set of colors.

Response 1.7

We agree with the Reviewer that the Seurat color scheme is not suitable for large samples, and thus we used our own color scheme (below) as it is presented in the top bars of heatmaps and other figures. We now show the UMAP plots representing each patient with the same color scheme.

1.8 Why do the proliferating CD8 T cells look like they express markers of all subtypes (Extended Data Fig. 3c again)? Are these cells merely doublets? To annotate them as proliferative, one must show that they preferentially express proliferation markers.

Response 1.8

Based on our use of more modest clustering thresholds on datasets from which low-quality samples were removed, we no longer have a proliferating CD8⁺ T cell cluster (revised Fig. 2a, Extended Data Fig. 4c).

1.9 Another problem with the clustering is that naïve-like and effector-like cells are present within the same cluster (for example, see the Memory-like Naïve and Proliferating clusters in Extended Data Fig. 3c). Again, this suggests that cells are clustering by technical variation or batch rather than by biology, at least to some extent.

Response 1.9

This has been solved with the new datasets that were clustered with the moderate resolution of clustering. We don't see the naïve or effector-like cells in the same cluster. The heatmap represents 7 distinct clusters by top DEGs (Extended Data Fig. 4c).

1.10 Figure 2b, c: only the GZMB-low GNLY+ cluster shows a significant difference between HCA and RA – the other cell types cited in the text are not convincingly different. Unfortunately, GZMB-low GNLY+ is the slightly dubious cell subtype with no clear distinguishing markers (see above).

Response 1.10

This has been solved with our expanded datasets and more modest clustering thresholds that now identify only one major cluster of GZMB+ GNLY+ CD8+ T cells. The new clusters based on more modest thresholds no longer have the GZMB^{low} GNLY+ cluster (Fig. 2a), and we now provide the expression level of GNLY and GZMB in revised Fig. 2c.

Revised Fig. 2a and Fig. 2c:

1.11 Figure 2d and f are based on the default DE gene analysis module in the Seurat package, which is not statistically valid because it assumes that each cell is a statistically independent replicate. There is a growing body of literature on this artifact and how to address it (see Nat Commun 2021; 12:738 and references therein). Again, this is a case where Seurat is popular because it gives you the result you want (in this case: spectacular DE gene p-values). In reality, the transcriptomes of cells within a single sample are highly correlated, so these p-values are misleading. A more statistically valid approach would be to construct the pseudobulk transcriptome of each sample and then perform DE analysis using DESeq or EdgeR (<https://www.biorxiv.org/content/10.1101/2022.02.16.480517v1>). This will give you very few DE genes if your cohort size is small. But they will be credible.

Response 1.11

We appreciate the Reviewer's suggesting this approach, and agree that the pseudobulk DE analysis is more accurate. In the revised manuscript, we computed the pseudobulk gene expression for each cluster of each sample using the scater R package¹. Thereafter, the pseudobulk expression counts were normalized and used for differential gene expression analysis of ACPA+ RA vs HC using DESeq2 R package². The calculated log₂ fold-change of gene expression in ACPA+ RA vs. HC multiplied by the -log (FDR) were used to rank the genes in descending order, following which gene-set enrichment analysis (GSEA) was performed for the GZMB+ cluster and the GZMK+ cluster. We then used the gene sets of gene ontology (GO) to identify the represented "biological process pathways". The normalized enrichment score (NES) and adjusted P value for multiple testing were used to define the enriched biological process pathways in either GZMB+ or GZMK+ clusters. The GO enrichment analysis revealed a

significant enrichment of T cell cytotoxicity and cell killing pathways in ACPA⁺ RA GZMB⁺ cluster (adjusted P < 0.05) compared to HC GZMB⁺ cluster, while the ACPA⁺ RA GZMK⁺ cluster pathways exhibited no significant difference(s) as compared to HC.

We updated the Results section in the manuscript, and updated Fig. 2d and 2e (directly below):

The pseudobulk expression of genes involved in cell killing and T cell cytotoxicity were visualized as a heatmap. Each column represents one sample. This heatmap is shown in updated Fig. 2f (directly below):

1.12 Also, it's not appropriate to use a raw p-value cutoff of 0.05 in DE gene analysis. There has to be some correction for multiple testing, since thousands of genes are being tested.

Response 1.12

We agree with the Reviewer, and now use the corrected P value, not the raw P value, to account for multiple testing in defining the DE genes.

1.13 From what I could tell, the GO term enrichment analysis in Fig. 2e, g compares DE genes to all genes in the genome. This is not a valid approach. DE genes need to be compared to the set of all genes expressed in CD8⁺ T cells, since a gene needs to be expressed to have a chance of being differentially expressed.

Response 1.13

In the GO analysis, we used the expressed genes in CD8⁺ T cells as a background list for the DE gene analysis, consistent with the approach previously described by others (<https://www.pnas.org/doi/epdf/10.1073/pnas.1722333115>)³.

1.14 Figure 3e indicates that the transcriptomes of cells from the same sample are indeed highly correlated with each other. For example, most of the RA cells expressing the gamma-delta T cell marker TRGV2 come from just 1 of the 9 RA samples. Most of the panels in Figures 2 and 3 ignore this profound sample-specificity. Based on the data analysis strategy used in these two figures, one would conclude that TRGV2 was uniformly upregulated in RA, but in fact it's only detectably upregulated in one single RA sample. Similarly, by ignoring sample-to-sample variation, the authors might conclude from Figure 3e that TRGV2 was a marker of all clonally expanded CD8+ T cells. But again, it's only a marker of clonally expanded cells from one RA sample. Many key markers discussed in this manuscript, including GZMB, GNLY, TBX21 and ZNF683, show similarly strong sample specificity in Figure 3e, which deepens the concern regarding the validity of naively aggregating all cells from all samples. To address this concern, most of the panels in these two figures need to be redone. The authors should confirm that their conclusions reflect trends that are consistent across samples rather than driven by one or two outliers. This would involve pseudobulk analysis, examination of the consistency of trends across RA samples, etc.

Response 1.14

In the current analysis, we identified a cluster of gamma-delta T cells (gd T cells). To confirm whether these cells are in fact gd T cells, we measured the percentage of all alpha-beta and gamma-delta T cell receptor (TCR) genes in each cell. The disruption of the TCR gene percentage showed that this gd T cell cluster possesses the lowest percentage of alpha-beta TCR genes and the highest percentage of gamma-delta TCR genes (see directly below).

% of ab-TCR gene counts

% of gd-TCR gene counts

From the below figures, it is clear that this cluster is not based on one sample. Nevertheless, we now provide the heatmap of gd TCR gene expression with sample IDs to further confirm that this cluster is not biased by one single sample. Also, using the pseudobulk gene expression, we confirm gd T cells are included in only cluster 5 in multiple different ACPA⁺ RA patients.

gd TCR cluster

Pseudobulk gene expression. Each dot represents one sample

1.15 Another concern after examining Figure 3e is that 80-90% of RA cells come from just 4/9 RA samples. Since the analyses here are mostly based on averaging across all cells, this implies that most of single cell conclusions are essentially derived from only 4 RA samples.

Response 1.15

We think this concern arose in part due to the low resolution of the color scheme in Fig. 3e. However, in the revised Fig. 3, we now add additional RA samples and use an improved color scheme to more effectively demonstrate that large clonal expansions are present in most ACPA+ RA samples (8/12 samples). In addition, for better visualization of the data, in revised Fig. 3e we split the heatmap into ACPA+ RA and healthy control groups.

Revised Fig. 3e:

1.16 Extended Data Fig. 5d reveals major technical confounders in the scRNA-seq data. 8-9 genes at the bottom of this heatmap are labeled as being associated with memory function, but the three genes at the bottom of this set (IL7R, FOS and JUN) form a coregulated module that is anti-correlated with the rest of the memory markers and in fact anti-correlated with most of the marker genes in this panel. These genes most likely represent stress response, rather than memory T cell function. Almost all cell clusters contain a subset of cells that strongly upregulate these three genes in unison. These three genes are often upregulated in response to the stresses cells face in the course of isolation and processing for scRNA-seq. I would suggest that the stressed cells be removed from the analysis, or at least analyzed separately.

Response 1.16

We do not present the analysis of RA only samples in the revised Figures because we now present an in-depth comparison between ACPA⁺ RA and HC using pseudobulk gene analysis and TCR composition analysis in the revised Figs. 2, 3 and revised Extended Data Figs. 4 and 5.

In an additional analysis (Fig. below), the purple-colored cluster of memory CD8⁺ T cells is defined as memory cells based on the expression of CXCR3⁺ CCR7^{low} CD45RA^{low}, as previously described [<https://www.nature.com/articles/44385.pdf>]⁴. We further show that this population is distinct from naïve CD8⁺ T cells that express high level of CD62L.

Moreover, we now performed QC analysis to remove results from dead and low-quality cells based on generally accepted parameters, including that the data from each sample contain: (i) <10% percent mitochondrial genes; (ii) > 500 total distinct mRNAs per cell [based on UMIs]; (iii) > 400 distinct genes per cell; and (iv) < 0.8 predicted cell doublets.

1.17 Even if one ignores the putative stress-responsive cells, the Central Memory and Memory-like Naïve clusters in Extended Data Fig. 5d show profound intra-cluster expression variation that's larger than the difference between these two clusters. In either cluster, one cell subgroup expresses naïve and memory markers, while another expresses cytotoxic effector markers. The same can be said of the “proliferating” cluster. Thus, a substantial fraction of the cells labeled as memory or memory-like or naïve may actually be effector cells.

Response 1.17

We appreciate the Reviewer's point, and now address this concern using expanded and more highly QC-controlled datasets as described in Response 1.1. Using our expanded and refined datasets, we re-clustered the CD8⁺ T cells and now observe distinct patterns of gene expression in naïve vs. memory vs. effector cells (revised Fig. 2a and Extended Data Fig. 4c).

1.18 More technical artifacts: in Extended Data Figure 5d, The “Naïve (i)” and “GZMG+ GNLY+ ZNF683+” clusters show a blurry vertical stripe, i.e. a patch of cells that don't show preferential expression of any particular marker. These are most likely low-quality cells, potentially derived from 1-2 low-quality samples. Perhaps those samples should be discarded.

Response 1.18

As described above in Response 1.1, we now address this concern through addition of data from 5 additional RA samples and by implementing more rigorous QC parameters that resulted in removal of 2 low-quality samples. Our revised heatmap clusters (below) no longer show a vertical stripe:

1.19 *Having read a bit further, I would say that my concerns regarding sample-specific effects vs trends shared across samples applies not just to Figures 2 and 3 but to most or all of the main and supplementary figures that are based on scRNA-seq data.*

Response 1.19

We updated our scRNA-seq cohorts by removing the 2 low-quality samples and adding additional 4 additional high-quality samples. The updated scRNA-seq datasets now include 12 ACPA+ RA and 6 healthy samples. The updated datasets and analysis address the concerns raised by the Reviewer.

1.20 *Despite all the data quality and data analysis concerns, the conclusion that “the large clonal expansions of CD8+ T cells are predominantly in the GZMB+ clusters as compared to GZMK+ clusters” does appear to be true. However, the word “predominantly” may be a bit too strong.*

Response 1.20

We appreciate the Reviewer’s comment. Revised Fig. 3b presents data on the distribution of large clonal expansions of CD8+ T cells in the GZMB+ as compared to GZMK+ clusters, and shows that the majority of these expansions are in the GZMB+ clusters. We removed the word “predominantly”.

Revised Fig. 3b:

1.21 Another major conclusion, namely that “CD8⁺ T cells in ACPA⁺ RA blood exhibit proliferative responses to cit-vimentin in a HLA class I-dependent manner,” also appears to be supported by the data (Figure 4).

Response 1.21

We appreciate the Reviewer’s comment. We demonstrated that citrullinated vimentin, but not native vimentin, specifically induces the proliferation of CD8⁺ T cells from ACPA⁺ RA patients in an HLA class I-dependent manner.

1.22 Figure 5: the conclusions are again very plausible, but again no attempt is made to determine if they are driven by one single atypical sample or if they represent a shared trend across all samples (cells are only analyzed in aggregate across all samples).

Response 1.22

We appreciate the Reviewer’s comment. We believe our conclusion from Fig. 5 represents a common feature of ACPA⁺ RA shared across multiple patients. As shown in Fig. 4c, we demonstrated that CD8⁺ T cells in 4 of 9 ACPA⁺ RA samples exhibit robust proliferation in response to cit-Vimentin, and that CD8⁺ T cells from 3 of 9 samples exhibit modest responses to cit-Vimentin. In Extended Data Fig. 16, we show that 10 of 13 ACPA⁺ RA patients possess CD8⁺ T cells reactive to cit-H3. Based on our results in Fig. 4, Fig. 5, and Extended Data Fig. 16, we conclude that the majority of ACPA⁺ RA samples contain CD8⁺ T cells specific for citrullinated antigens.

1.23 It’s not clear what one could conclude from Figure 5g, h. These panels show that there are cells in the synovium that are transcriptomically similar to the GZMB and GZMK-expressing populations in peripheral blood. But the degree of similarity is not obvious. Also, these results don’t tell us anything about whether or not the GZMB-expressing synovial cells are clonally expanded. What is the message here? Do we need these two panels?

Response 1.23

We have now performed new analysis on paired RA blood and synovial tissue samples, and show that the same clonally expanded GZMB⁺ GNLY⁺ CD8⁺ T cell lineages in the blood are also present and expanded in the synovium. The majority of the clonally expanded CD8⁺ T cells express GZMB and GNLY, and in contrast only a few CD8⁺ T cells and lineages express GZMK. We thereby

demonstrate that these *GZMB*⁺ *CD8*⁺ T cells are clonally expanded in synovium (Revised Fig. 5g,h and Revised Extended Data Figs. 12 and 13), can be activated by citrullinated antigens to mediate cell killing (Extended Data Fig. 17), and thereby suggest they could mediate synovial tissue destruction in RA.

Revised Fig. 5g and h:

Revised Extended Data Fig. 12:

Revised Extended Data Fig. 13:

1.24 The responsiveness of RA CD8⁺ T cells to citrullinated vs native vimentin also appears to be a robust conclusion (Figure 6).

Response 1.24

We appreciate the Reviewer's comment, and agree that citrullinated vimentin and other citrullinated antigens specifically activated ACPA⁺ RA CD8⁺ T cells as compared to their native forms.

Minor comments

1.25 scRNA data: how many single cells were analyzed from each sample? Were there any low-quality samples that should have been discarded? Or any samples that contributed a negligible number of cells, too few to discern any meaningful trend? I suspect the answer is yes to the latter question.

Response 1.25

The number of CD8⁺ T cells varied across the samples, ranging from a few hundred to several thousand. We have now removed the data from all samples with < 100 CD8⁺ T cells. Below is the distribution of CD8⁺ T cell numbers across the samples included in our updated datasets:

1.26 *The term CCP is first used on pg. 4 but defined only on pg. 12.*

Response 1.26

We have now added the short description of CCP in the Introduction section (Page 3):
 “The detection of ACPA in serum using the anti-cyclic citrullinated peptide (CCP) test, is a well-established component of the diagnostic criteria for seropositive RA.”

1.27 *“CD8A >0.5 and CD4 <0.1” – in what units are the expression levels quantified?*

Response 1.27

Expression levels were quantified based on the log Normalized counts measured by Seurat, and this information is provided in the Methods section.

1.28 *Extended Data Fig. 3: what does “fold change>0.8” mean? Is this a log-fc? Log2? Log10? There are multiple places in the manuscript where numbers are given without explaining the units or how they were calculated.*

Response 1.28

The fold change is a log₂ fold change, and is now detailed in the Figure legend section (Revised Extended Data Fig. 4).

1.29 *Extended Data Fig. 3c: please use the same cluster colors as in Fig. 2a.*

Response 1.29

We now use matched cluster colors.

Revised Fig. 2a and Extended Data Fig. 4c:

1.30 The Methods section states that the authors initially clustered all cells and then, in a second round, clustered only CD8+ T cells. I could only find figures and results from the latter. Was the all-cell clustering result ever used in the manuscript? And if it was not used, why was it done?

Response 1.30

The all CD3+ T cell cluster was initially performed to identify the CD8+ T cell cluster, based on the T cells in this cluster expressing CD8A mRNA (by 10X gene expression) and CD8A protein (by 10X CITE-Seq) – see Figure below. Potential cell doublets were identified using the scds R package (<https://doi.org/10.1093/bioinformatics/btz698>)⁵, and the droplets/cells with doublet scores > 0.8 were removed.

Clustering approach used to identify CD8⁺ T cells:

1- Cluster all CD3⁺ T cells

2- Remove potential doublet cells

Clusters 2, 7, 9 and cells with doublet score > 0.8

3- Select CD8⁺ T cell clusters

Clusters 3, 4, 5

4- Re-cluster CD8⁺ T cells

1.31 CD8⁺ T cells in scRNA-seq data were defined based on these FACS-like criteria: CD3E>0.5 & CD8A>0.5 & CD4<0.1. This is not a reliable way of analyzing scRNA-seq data, which have higher noise and dropouts than FACS data. The more conventional way to identify cell types is to cluster the single cell transcriptomes and map each cluster (or group of clusters) to a cell type or subtype based on markers. CD8⁺ T cells should be identifiable in this manner.

Response 1.31

We agree, and in our manuscript use the approach described by the Reviewer. As detailed in our Response to 1.30, we first clustered all CD3⁺ T cells, and then identified CD8⁺ T cells based on their cluster expressing both CD8A mRNA (by 10X gene expression) and CD8A protein (by 10X CITE-Seq) – as illustrated in the Figure in Response 1.30.

1.32 How was “batch” defined in the batch correction step? Was each sample a batch?

Response 1.32

In a batch correction, we considered each sample as a batch.

1.33 Figure 2c: the pvalue is apparently obtained from two-way ANOVA. Which two factors were considered in this two-way analysis? The bar graph should be replaced by a figure showing the actual distribution of data points. There are only 14 data points, so it should be possible to show all of them as dots. This way, we can determine if the ANOVA results are credible or if they are driven by one or two outlier data points (ANOVA assumes a normal distribution, so one or two outliers can trigger a strong p-value).

Response 1.33

The bar graph and ANOVA analyses have been removed.

1.34 Similarly, Figure 2h should show the expression level in each of the 14 samples and the De gene p-value should be specified for each gene, so we can tell if the expression difference is significant.

Response 1.34

We replaced this figure with a revised version based on pseudobulk DE analysis. The pseudobulk expression of genes included in the core enrichment score for cell killing and T cell cytotoxicity were visualized as a heatmap. Each column represents one sample.

Revised Fig. 2f:

The expression of key-genes is presented below, and includes a P value calculated using the Wilcox test. Each dot represents the pseudobulk expression of one sample.

Revised Fig. 2g:

1.35 GNLY is one of the major markers used to label CD8+ cell subtypes - it should be shown in Figure 2b.

Response 1.35

We agree with the Reviewer's comment. Revised Fig. 2c shows the expression of the key markers including GNLY:

Revised Fig. 2c:

1.36 The umaps in Figures 2a and 3a should be the same(?).

Response 1.36

Prior Fig. 3a was another version of the UMAP. Updated Figs. 2a and 3a now use the same UMAP plot.

Revised Figs. 2a and 3a:

1.37 Figure 3e is more useful than some of the other heatmaps, because disease status and sample ID are indicated in the top bar. It would be useful to show these covariates in the other heatmaps as well. Plus QC metrics such as mitochondrial read fraction and number of features.

Response 1.37

We updated the heatmaps with colored bars to show the most important covariates including mitochondria gene %, read count, number of features, and frequency of TCR clones in revised Fig. 3e:

Reviewer #2 (Remarks to the Author):

The paper by Moon et al investigate the presence and function of CD8+ T cells in Rheumatoid Arthritis and the correlation with ACPA positive disease. The study focusses on T cells from blood and define a subpopulation of GZMK/CD8+ population. Further they study the activation by citrullinated antigens. I believe the study is interesting but have some concerns:

Major issues:

2.1 *The study also includes data on $\gamma\delta$ CD8+ cells that are thought to not be MHC restricted at least not to recognize peptides like other CD8+ T cells. What is their role here and since they are expanded are they affecting the data with killing that in part is "MHC independent"? I find this data curious, and it should be discussed both in the introduction and discussion part. Here they are lumped with CD8+ cells in general which might not be adequate.*

Response 2.1

The Reviewer asks an important question about the potential role of $\gamma\delta$ CD8⁺ T cells in RA, and if the population can affect the MHC-independent killing effect of CD8⁺ T cells in response to citrullinated antigens. Previous studies on $\gamma\delta$ T cells have mainly focused on the subpopulation of CD4⁻CD8⁻ $\gamma\delta$ T cells have a restricted T cell receptor (TCR) and their TCRs are non-MHC restricted⁶. A few reports demonstrated that a small population of $\gamma\delta$ T cells express CD8 and exhibit a high cytotoxic potential based on their expression of IFN γ and TNF α ⁷. Gaballa et al. showed TCR stimulation induced the activation and proliferation of $\gamma\delta$ CD8⁺ T cells, suggesting $\gamma\delta$ CD8⁺ T cells possess adaptive rather than simply innate phenotypes⁸. Also, in rheumatoid arthritis, it has been reported $\gamma\delta$ T cells are expanded and activated by synthetic alkyl phosphates, isopentenyl pyrophosphate, or HSP65^{9,10}, suggesting $\gamma\delta$ T cells in RA may recognize synovial antigens and be stimulated to express inflammatory mediators.

Based on our data, we cannot determine with certainty if the $\gamma\delta$ CD8⁺ T cells can mediate MHC-independent killing in response to citrullinated antigens. We expect $\gamma\delta$ CD8⁺ T cells in RA patients might act as killer cells like traditional CD4⁻CD8⁻ $\gamma\delta$ T cells, however we believe the potential killing activity of $\gamma\delta$ CD8⁺ T cells would only minimally impact the overall MHC-independent killing activity of all CD8⁺ T cells due to the proportion of $\gamma\delta$ CD8⁺ T cells (< 5% in CD8⁺ T cells) being considerably lower than that of $\alpha\beta$ CD8⁺ T cells. Also, the MHC class I-independent cytotoxicity of CD8⁺ T cells in ACPA⁺ RA blood was induced in response to citrullinated antigens, while native antigens did not induce cytotoxic activity of CD8⁺ T cells (Extended Data Fig. 17). It is possible that this MHC-independent cytotoxicity might be mediated by release of cytolytic mediators (Granzymes and CD107a). Further, when the interaction of MHC class I of antigen-presenting cells and TCR of CD8⁺ T cells was blocked by anti-HLA class I blocking antibody, the proliferation and activation of CD8⁺ T cells were almost completely inhibited (Figure 4 and 6). These results suggest that the cytotoxic activity of CD8⁺ T cells is in part dependent on MHC class I, even though we used MHC-unmatched CD8⁺ T cells and cancer cell line in the cytotoxicity assay (Extended Data Fig. 17c,d). In conclusion, further investigation is needed to better define the potential role of MHC class I-dependent CD8⁺ T cell activation and cytotoxicity in RA, and if $\gamma\delta$ CD8⁺ T cells in ACPA⁺ RA can be activated by citrullinated antigens to mediate cytotoxic killing.

To address these possibilities, we added the following text regarding the potential role of $\gamma\delta$ CD8⁺ T cells the Discussion:

Discussion:

“It is possible that the HLA class I-independent killing activity of ACPA⁺ RA CD8⁺ T cells might be mediated by an expanded subpopulation of TCR $\gamma\delta$ ⁺CD8⁺ T cells, even though this subpopulation represents a low proportion of the CD8⁺ T cells in ACPA⁺ RA (<5% of total CD8⁺ T cells as shown in Fig. 1e). It has been reported that TCR $\gamma\delta$ ⁺CD8⁺ T cells secrete pro-inflammatory and cytotoxic molecules like conventional TCR $\gamma\delta$ ⁺CD4⁻CD8⁻ T cells, and that this subpopulation could act as adaptive immune cells that are activated and mediate cytotoxic killing in response to specific antigens. Nevertheless, further investigation is needed to more fully determine if the TCR $\gamma\delta$ ⁺CD8⁺ T cells observed in ACPA⁺ RA are specific for and activated by citrullinated antigens.”

2.2 *It has been known for some time that ACPA positivity correlate with CD8+ T cells numbers (for example: de Hair 2013). This data needs to be referenced and other mechanisms discussed, such as the study by Jung et al that showed recently that CD8 T cells are present in the synovium and their activation is coupled with citrullination of antigen. They also show TCR repertoire and skewing of this. To me this is more reasonable as Antibodies are coupled to CD4 T cell activation which typically balance a CD8 response. Thus, one would expect more CD8 T cell activation in the ACPA- group of patients.*

Response 2.2

We agree with the Reviewer's comment that we need to include references and descriptions of previous findings related to the correlation of ACPA positivity and CD8⁺ T cell responses in RA. Our manuscript provides important advances in further defining the specific subpopulations of CD8⁺ T cells that are present, expanded, and reactive with citrullinated antigens in ACPA⁺ RA

The reports from de Hair *et al.*¹¹ and Jung *et al.*¹² were previously and are currently cited in our manuscript.

de Hair *et al.* describe the association of ACPA and the presence of CD8⁺ T cells in synovium in RA. This study analyzes the synovia of 55 preclinical high-risk individuals who are rheumatoid factor and ACPA positive by performing proportional hazards regression analysis, and found that the expression of CD8 in the synovium is associated with arthritis development. Further, they show that the presence of CD8⁺ T cells in the synovium is correlated with ACPA reactivity to citrullinated vimentin or fibrinogen, suggesting ACPA⁺ RA synovial CD8⁺ T cells may be activated by specific citrullinated protein and that these CD8⁺ T cells might be involved in early stage of RA onset. Nevertheless, they do not define the subpopulations of CD8⁺ T cells in blood and synovium, and do not assess if the CD8⁺ T cells in RA are specific for and activated by citrullinated antigens. Our findings provide important new insights through an in-depth analysis of ACPA⁺ RA CD8⁺ T cells using single cell transcriptomics and TCR repertoire sequencing, combined with *in vitro* characterization of citrullinated antigen-reactivity and cytotoxicity.

Jung *et al.* detect CD69⁺CD103^{+/-}CD8⁺ T cells in synovial fluid mononuclear cells (SFMC) using flow cytometry and CDR3 sequencing. They demonstrate that RA SF CD69⁺CD103^{+/-}CD8⁺ T cells exhibit effector memory tissue-resident phenotypes and oligoclonal expansion. Moreover, the cells can be activated by IL-15 and induce histone citrullination with neutrophil NET formation in a perforin-dependent manner, indicating CD69⁺CD103^{+/-}CD8⁺ T cells may be an important population in RA pathogenesis. However, this study also does not show the reactivity against citrullinated antigens and no transcriptomic data is shown.

In summary, while previous studies advanced our understanding of the involvement of CD8⁺ T cells in RA pathogenesis, our study provides key insights through an in-depth analysis that shows CD8⁺ T cell specific responses and clonal expansion in response to citrullination antigens, thereby providing a plausible mechanism for CD8⁺ T cell activation and possibly cytotoxic tissue destruction in ACPA⁺ RA.

2.3 *Much of the data is shown as proportions or percentages. For some of the key data it would be nice with absolute numbers.*

Response 2.3

We appreciate the Reviewer's suggestion, and have now added the absolute number of cells for the key data in Extended Data Fig. 3a-e using Precision Count Beads (BioLegend). This includes total CD8⁺ T cells, granzyme B-expressing CD8⁺ T cells, KIR2DL3-expressing CD8⁺ T cells, and CD8⁺ T cell memory subsets, in the healthy control, ACPA⁻ RA, and ACPA⁺ RA PBMC samples studied. The comparison of absolute cell numbers between each group is similar in terms of the proportions of the subpopulations. Total or granzyme B or KIR2DL3-expressing CD8⁺ T cell numbers in ACPA⁺ RA blood are significantly higher than those in healthy controls, and granzyme K⁺ CD8⁺ T cell numbers in blood are similar between all groups. Also, ACPA⁺ RA CD8⁺ T cells exhibit increased frequencies of the TEMRA subset as compared to healthy controls.

To incorporate these data, we made the following revisions to the manuscript:

Main text:

"Further, we found that not only the proportion of total CD8⁺ T cells, GzmB-expressing, KIR2DL3-expressing, or EMRA CD8⁺ T cell subsets, but also the absolute cell counts of these activated cytotoxic CD8⁺ cells, are increased in ACPA⁺ RA blood as compared to HC (Extended Data Fig. 3)."

Extended Data Fig. 3:

Revised “Extended Data Fig. 3: Absolute cell counts of CD8⁺ T cells in ACPA⁺ RA patients. Absolute cell numbers of total CD8⁺ T cells (a), granzyme B-expressing CD8⁺ T cells (b) or granzyme K-expressing CD8⁺ T cells (c), memory CD8⁺ T cells (d) or KIR2DL3⁺CD8⁺ T cells (e) per 1 μl assessed by counting beads in HC (n = 6 or 8), ACPA⁻ (n = 5 or 7) or ACPA⁺ (n = 6, 7 or 8) RA PBMCs. Data are presented as means ± SEM. *P < 0.05, **P < 0.01 or ***P < 0.001 by one-way ANOVA (a-e). ns, not significant.”

Methods:

“To count absolute cell numbers of each population, Precision Count Beads (BioLegend) were added to the stained cells and samples analyzed by flow cytometry. The absolute cell number was calculated based on the following formula: absolute cell count (cell/μl) = (cell count x Precision Count Beads volume) / (Precision Count Beads count x cell volume) x Precision Count Bead concentration.”

2.4 In figure 1F the data for ACPA- RA should be shown.

Response 2.4

We thank the Reviewer for this suggestion. We have now included the data for the proportion of Granzyme B⁺ or Granzyme K⁺ CD8⁺ T cells in ACPA⁻ RA blood in Figure 1f with representative flow cytometry data and graphs (See updated Fig. 1f). There are no significant differences between ACPA⁻ RA and ACPA⁺ RA or healthy controls in Granzyme B or Granzyme K-expressing CD8⁺ T cell populations.

Revised Fig. 1f:

2.5 In Figure 4 the proliferation to citrullinated antigen needs a control. Either HC or ACPA⁻ RA or both.

Response 2.5

We appreciate that there was no control for the proliferation assay in response to citrullinated antigens in Figure 4. We have repeated the PBMC stimulation assay using PBMCs of ACPA⁻ RA patients and healthy controls and analyzed the levels of Ki-67 in CD8⁺ T cells (See updated Extended Data Fig. 7a). Citrullinated Vimentin did not induce Ki-67 expression in both ACPA⁻ RA and HC CD8⁺ T cells. Additionally, citrullinated antigens did not stimulate CD8⁺ T cells in ACPA⁻ RA and HC to proliferate (Extended Data Fig. 7b,c).

We amended the main text and Figure legend:

Main text:

“In contrast, cit-vimentin did not induce proliferation of ACPA⁻ RA or HC CD8⁺ T cells (Extended Data Fig. 7a-c).”

Extended Data Fig. 7:

“Extended Data Fig. 7: Citrullinated vimentin does not induce the proliferation of ACPA⁻ RA or HC CD8⁺ T cells. a, Percentage of Ki-67⁺CD8⁺ T cells in ACPA⁻ RA ($n = 5$) or HC ($n = 7$) PMBCs after stimulation with anti-CD3/28 antibodies, NP (Influenza)/ pp65 (CMV) proteins (50 μM of each), citrullinated vimentin (100 μM), or native vimentin (100 μM) for 16 hr. **b,c,** Quantification of the proliferating CD8⁺ T cells in co-culture of MoDCs with ACPA⁻ RA or HC total CD3⁺ T cells (b, $n = 4$ or 5) or CD8⁺ T cells only (c, $n = 4$). Bars represent means ± SEM. * $P < 0.05$ or *** $P < 0.001$ by ordinary one-way ANOVA with Tukey’s multiple comparison test.”

Minor issues:

2.6 The manuscript needs to be edited for clarity. Some sentences are very long and there is some run on.

Response 2.6

We have carefully reviewed and edited the manuscript to make it more clear, and eliminated several of the very long sentences.

Reviewer #3 (Remarks to the Author):

This is an interesting study where the authors have investigated the CD8 T cell response to citrullinated self antigens in rheumatoid arthritis where CD4 T cells are more commonly associated. The study has been well performed and includes important controls such as ACPA negative patients. The results demonstrate that CD8 T cells are clonal expanded and activated in RA, have relevant cytotoxic and joint homing markers as well as the capacity to kill target cells in an in vitro assay. The authors have been careful and circumspect in their language as they are unable to demonstrate a direct causative link between CD8 T cells, their activation and pathology in RA patients. Overall, the work highlights the potential contribution of these cells and therefore is a relevant contribution. It would perhaps be important to examine a number of additional things to demonstrate cause and effect.

3.1 Responses to irrelevant citrullinated peptides.

Response 3.1

We appreciate the Reviewer's comments. Our study uses only recombinant whole proteins including citrullinated and native (arginine) forms, not peptides. For CD4⁺ T cells, multiple studies have reported citrullinated epitopes binding to HLA-DRB1 and stimulating CD4⁺ T cells in RA blood¹³. In this study, we demonstrate for the first time reactivity of CD8⁺ T cells in ACPA⁺ RA to citrullinated antigens. Further studies are needed to define the specific citrullinated peptides that bind to HLA class I and elicit cytotoxic CD8⁺ T cell responses in RA. We include native forms of the citrullinated antigens as negative controls, instead of using irrelevant citrullinated antigens. In multiple experimental systems, we confirmed that native proteins do not stimulate the activation or proliferation of CD8⁺ T cells in ACPA⁺ RA blood (Figures 4, 5 and 6).

3.2 In vitro and/or in vivo demonstration of pathological effects of CD8 T cells on joint cells using e.g. organoids, tissues slice or humanised murine models.

Response 3.2

We agree with the Reviewer that we need to demonstrate the pathological effects of CD8⁺ T cells in RA joints. Previous reports have found that CD8⁺ T cells migrate and accumulate in RA synovium, and that in the synovium they have a phenotype of resident-memory T cells (T_{RM}) producing pro-inflammatory cytokines and cytotoxic mediators based on immunohistochemical staining of RA synovial tissue sections or RNA sequencing of synovium-derived cells¹⁴. Margaret *et al.* demonstrate that synovial T_{RM} are activated in RA joint flares, and that depletion of synovial T_{RM} inhibits inflammatory arthritis in mice, suggesting CD8⁺ T cells contribute to joint destruction in RA pathogenesis. However, there are limitations that *in vitro* or humanized mouse models do not fully reflect the complex micro-environment of RA synovium. We have indirectly demonstrated the cytotoxic potential of self-antigen reactive CD8⁺ T cells using *in vitro* cell killing assays. Even though this cytotoxic activity was HLA-independent, we anticipate that citrullinated antigens activate CD8⁺ T cells to release cytotoxic and lytic mediators including perforin, granzymes and IFN γ that contribute to joint tissue inflammation and destruction in RA.

We appreciate that our conclusions regarding the potential pathological role of CD8⁺ T cells in RA was unclear, and revised the Discussion to address this:

Discussion:

“In RA, it is well established that there are increased levels of CD8⁺ T cells in synovium, including tissue-resident memory CD8⁺ T cell populations that potentially contribute to joint destruction by releasing pro-inflammatory cytokines in response to antigens. Here, we showed that ACPA⁺ RA CD8⁺ T cells mediate cytotoxic responses and kill DLD-1 tumor cells by releasing cytolytic granules in response to cit-vimentin or cit-H3, but not their native forms. Even though this cytotoxic activity occurred in an HLA-independent manner using an HLA-unmatched cancer cell line, this HLA-independent cytotoxicity was mediated by citrullinated antigen-activated ACPA⁺ RA CD8⁺ T cell lytic responses. Our results suggest that in ACPA⁺ RA, synovial CD8⁺ T cells could be activated by citrullinated antigens to mediate joint tissue destruction.”

References

1. McCarthy, D.J., Campbell, K.R., Lun, A.T. & Wills, Q.F. Scater: pre-processing, quality control, normalization and visualization of single-cell RNA-seq data in R. *Bioinformatics* **33**, 1179-1186 (2017).
2. Love, M.I., Huber, W. & Anders, S. Moderated estimation of fold change and dispersion for RNA-seq data with DESeq2. *Genome Biol.* **15**, 1-21 (2014).
3. Younis, S. et al. Multiple nuclear-replicating viruses require the stress-induced protein ZC3H11A for efficient growth. *Proc. Natl. Acad. Sci. U.S.A.* **115**, E3808-E3816 (2018).
4. Sallusto, F., Lenig, D., Förster, R., Lipp, M. & Lanzavecchia, A. Two subsets of memory T lymphocytes with distinct homing potentials and effector functions. *Nature* **401**, 708-712 (1999).
5. Bais, A.S. & Kostka, D. scds: computational annotation of doublets in single-cell RNA sequencing data. *Bioinformatics* **36**, 1150-1158 (2020).
6. Lawand, M., Déchanet-Merville, J. & Dieu-Nosjean, M.-C. Key features of gamma-delta T-cell subsets in human diseases and their immunotherapeutic implications. *Front. Immunol.* **8**, 761 (2017).
7. Kadivar, M., Petersson, J., Svensson, L. & Marsal, J. CD8ab gd T Cells: A Novel T Cell Subset with a Potential Role in Inflammatory Bowel Disease. (2016).
8. Gaballa, A., Arruda, L., Rådestad, E. & Uhlin, M. CD8+ $\gamma\delta$ T cells are more frequent in CMV seropositive bone marrow grafts and display phenotype of an adaptive immune response. *Stem cells international* **2019**(2019).
9. Laurent, A.J., Bindslev, N., Johansson, B. & Berg, L. Synergistic Effects of Ethanol and Isopentenyl Pyrophosphate on Expansion of $\gamma\delta$ T Cells in Synovial Fluid from Patients with Arthritis. *Plos one* **9**, e103683 (2014).
10. Tanaka, Y. et al. Nonpeptide ligands for human gamma delta T cells. *Proc. Natl. Acad. Sci. U.S.A.* **91**, 8175-8179 (1994).
11. De Hair, M. et al. Features of the synovium of individuals at risk of developing rheumatoid arthritis: implications for understanding preclinical rheumatoid arthritis. *Arthritis Rheumatol.* **66**, 513-522 (2014).
12. Jung, J.H. et al. Synovial fluid CD69+ CD8+ T cells with tissue-resident phenotype mediate perforin-dependent citrullination in rheumatoid arthritis. *Clinical & translational immunology* **9**, e1140 (2020).
13. Law, S.C. et al. T-cell autoreactivity to citrullinated autoantigenic peptides in rheumatoid arthritis patients carrying HLA-DRB1 shared epitope alleles. *Arthritis Res. Ther.* **14**, 1-12 (2012).
14. Chang, M.H. et al. Arthritis flares mediated by tissue-resident memory T cells in the joint. *Cell reports* **37**, 109902 (2021).

REVIEWER COMMENTS

Reviewer #1 (Remarks to the Author):

The authors have done a truly outstanding job of revising the data analysis and the figures to fix all the issues I raised. This is rigorous, high-quality science, and an important advance in RA immunology.

Reviewer #2 (Remarks to the Author):

I believe that the manuscript has improved allot upon editing and support publication.

Reviewer #3 (Remarks to the Author):

The authors have adequately addressed my concerns